# Self-Exploring Language Models for Explainable Link Forecasting on Temporal Graphs via Reinforcement Learning

## Abstract

Forecasting future links is a central task in temporal graph (TG) reasoning, requiring models to leverage historical interactions to predict upcoming ones. Traditional neural approaches, such as temporal graph neural networks, achieve strong performance but lack explainability and cannot be applied to unseen graphs without retraining. Recent studies have begun to explore using large language models (LLMs) for graph reasoning, but most of them are constrained to static graphs or small synthetic TGs and lack the evaluation of the quality of reasoning traces generated by LLMs. In this work, we present **Rea**soning-Enhanced **L**earning for **T**emporal **G**raphs (ReaL-TG), a reinforcement learning framework that fine-tunes LLMs to perform explainable link forecasting on real-world TGs. ReaL-TG uses outcome-based reward to encourage models to self-explore reasoning strategies from graph structure and to produce explanations that directly justify their predictions. To enable evaluation on LLM-generated reasoning traces, we propose a new evaluation protocol combining ranking metrics with an LLM-as-a-Judge system that assesses both the quality of reasoning and the impact of hallucinations. Experiments with ReaL-TG-4B, obtained by fine-tuning Qwen3-4B under our framework, show that it outperforms much larger frontier LLMs, including GPT-5 mini, on ranking metrics, while producing high-quality explanations confirmed by both the LLM judge and human evaluation.

## 1 Introduction

Temporal graphs (TGs) represent node interactions as links annotated with timestamps (Kazemi et al., 2020), making them well-suited for modeling a wide range of real-world scenarios such as social and transaction networks (Huang et al., 2023). This expressiveness has fueled the growing interest in TG reasoning, which focuses on capturing the dynamic graphical structures within TGs to support various downstream tasks. A widely studied task in TG reasoning is future link prediction, also known as link forecasting. It aims to predict future interactions between nodes based on historical node interactions, which is particularly useful in practical applications such as recommendation systems (Fan et al., 2021), community discovery (Rossetti & Cazabet, 2018) and financial analysis (Shamsi et al., 2022). Mainstream methods for link forecasting train neural-based models such as temporal graph neural networks (TGNNs) (Xu et al., 2020; Ma et al., 2020; Wang et al., 2021b; Gravina et al., 2024), memory networks (Rossi et al., 2020; Liu et al., 2022a), and sequence modeling units (Yu et al., 2023; Tian et al., 2024; Ding et al., 2025) on the training set of a TG, and then apply the trained model to the test set of the same TG. While effective, they suffer from two key limitations. First, most neural-based models lack human-readable explanations for their predictions, making it difficult for users to assess the trustworthiness of the results. Second, they typically require retraining when adapted to a new TG, and therefore cannot seamlessly generalize to unseen graphs.

Recently, the rapid scaling of language models has made them increasingly effective at generating coherent text, leading to their widespread adoption in question answering (QA) tasks across diverse domains. Building on this progress, an emerging line of research investigates whether large language models (LLMs) can also reason over graphs by prompting them to answer graph-related (such as link prediction) questions. Compared with traditional graph reasoning methods, LLMs naturally provide human-readable explanations and exhibit strong zero-shot generalization, suggesting the

potential to handle previously unseen graphs without retraining. Nevertheless, most existing studies concentrate on static graphs (Chai et al., 2023; Perozzi et al., 2024; Fatemi et al., 2024; Chen et al., 2024; Guo et al., 2025), and only a few have investigated TGs. Among these, several efforts focus on TGs with textual attributes and demonstrate strong performance (Lee et al., 2023; Liao et al., 2024; Wang et al., 2024; Wu et al., 2025). However, such settings carry a risk of data leakage, since textual features—including those directly relevant to prediction and even the correct answers to the questions—may already have been seen during pre-training (Ding et al., 2024). In contrast, LLM4DyG (Zhang et al., 2024b) evaluates LLMs on TG reasoning using fully synthetic graphs anonymized from text, thereby avoiding leakage. Yet its experiments are restricted to very small scales (up to 20 nodes), limiting the applicability of the findings to realistic scenarios. Moreover, existing studies largely overlook the evaluation of LLMs' reasoning outputs. Strong performance on link prediction metrics such as accuracy does not necessarily imply that the underlying reasoning traces are correct. In practice, LLMs may generate flawed reasoning or introduce hallucinations that still lead to the right prediction label, raising concerns about their reliability.

Building on these observations, we propose **Rea**soning-Enhanced **L**earning for **T**emporal **G**raphs (ReaL-TG), a reinforcement learning (RL) framework that fine-tunes LLMs to do perform link forecasting over TGs. Unlike prior works that rely on textual attributes or synthetic toy datasets, Real-TG is developed and evaluated on anonymized real-world TGs (where nodes are represented with numerical IDs without any semantic information) provided by the popular Temporal Graph Benchmark (TGB) (Huang et al., 2023), making it both practical and aligned with real application needs. By removing semantic information from textual attributes, anonymized graphs prevent data leakage and require the model to reason solely over the temporal graphical structures, leading to reasoning patterns focusing on the intrinsic dynamics of TG evolution. During RL, we choose a reasoning LLM, i.e., Qwen3 (Yang et al., 2025), as the base model and adopt Grouped Regularized Policy Optimization (GRPO) (Shao et al., 2024) together with an outcome-based reward tailored to TG link forecasting. This outcome-based setup not only encourages the model to self-explore reasoning strategies through its own textual outputs without process-level supervision, but also compels it to produce human-readable explanations that justify its predictions. In this way, the model is pushed to achieve both strong predictive accuracy and logically sound reasoning that supports its answers. To comprehensively evaluate LLMs in TG link forecasting, we further propose a new evaluation protocol tailored to this setting. First, we formulate the task as QA, where an LLM must directly generate the set of nodes it predicts as correct answers. On top of this formulation, we introduce penalized mean reciprocal rank (pMRR), an extension of MRR (Voorhees & Tice, 2000) that discounts the score when predicted nodes fall outside the ground-truth set, thereby discouraging over-generation. Second, to assess the quality of LLM-generated reasoning traces , we design an LLM-as-a-Judge (Zheng et al., 2023) evaluation with three criteria: (i) faithfulness, whether the reasoning is supported by the input graph; (ii) logical consistency, whether the reasoning follows a coherent and valid chain; and (iii) answer–explanation alignment, whether the predicted answers are justified by the model's own reasoning.

We summarize our contributions as follows:

- We propose ReaL-TG, the first framework that enables LLMs to perform explainable and effective link forecasting on real-world temporal graphs via reinforcement learning.
- We introduce a new evaluation protocol for TG link forecasting with LLMs that assesses not only prediction accuracy but also reasoning quality and the impact of hallucinations.
- Our fine-tuned model ReaL-TG-4B outperforms much larger frontier LLMs on both seen and unseen graphs. In addition, it produces high-quality explanations, as confirmed by both the LLM judge and human evaluation.

## 2 RELATED WORK & PRELIMINARIES

### 2.1 RELATED WORK

**Traditional Link Forecasting Methods.** Traditional approaches to TG link forecasting span several modeling paradigms. Memory-based methods such as TGN (Rossi et al., 2020) and TNCN (Zhang et al., 2024a) maintain evolving node memories to capture temporal dynamics, often combined with a Graph Neural Network (GNN) to aggregate graph information. Another line of works,

including JODIE (Kumar et al., 2019), TCL (Wang et al., 2021a), DyGFormer (Yu et al., 2023), and DyGMamba (Ding et al., 2025), leverages sequence modeling units such as recurrent neural networks, Transformers (Vaswani et al., 2017), and Mamba layers (Gu & Dao, 2023) to model temporal dynamics. Heuristic-based approaches like EdgeBank (Poursafaei et al., 2022) and Base 3 (Kondrup, 2025) avoid learnable parameters altogether, instead relying on carefully designed algorithms to extract relevant information from past interactions. Pure MLP-based methods such as GraphMixer (Cong et al., 2023) have also shown promise by directly encoding link information. Finally, snapshot-based methods like ROLAND (You et al., 2022) and UTG (Huang et al., 2024) adapt standard GNN architectures to TGs by modifying their training and inference procedures. While effective on standard benchmarks, these methods require retraining from scratch (often with hyperparameter tuning) when applied to new datasets, and they provide no explanations for their predictions, limiting their applicability in settings where interpretability is critical.

**LLMs for Graph Reasoning.**  A growing body of research explores LLMs' reasoning abilities on graph-related tasks. Fatemi et al. (2024) show that appropriate graph encodings can improve performance. Methods such as GraphToken (Perozzi et al., 2024), GraphLLM (Chai et al., 2023), and LLaGA (Chen et al., 2024) enhance reasoning by jointly training LLMs with graph representations, while G1 (Guo et al., 2025) further demonstrates that RL improves reasoning on static graphs. Recent works have started to examine LLMs' capabilities on TGs. LLM4DyG (Zhang et al., 2024b) shows that LLMs capture basic spatio-temporal dependencies but struggle with multi-hop reasoning, and its evaluation is limited to small synthetic TGs. Li et al. (2025) explore in-context learning (ICL) on TGs, showing that performance is highly sensitive to prompt design and subgraph selection. Concurrently, TGTalker (Huang et al., 2025b) investigates ICL-based link forecasting on real-world TGs. Despite these advances, none of the existing works addresses how to systematically evaluate LLMs' reasoning quality or how to guide them, through training, towards more effective reasoning strategies for link forecasting on real-world TGs.

## 2.2 Preliminaries

We first define TG as follows. Note that, in this work, we deliberately exclude node and edge features, focusing instead on how LLMs can reason over TGs solely from their topological structure.

**Definition 1 (Temporal Graph)** *Let $\mathcal{N}$ and $\mathcal{T}$ denote a set of nodes and timestamps, respectively. A TG can be represented as a sequence of $|\mathcal{G}|$ chronological interactions $\mathcal{G} = \{(u_i, v_i, t_i)\}_{i=1}^{|\mathcal{G}|} \subseteq \mathcal{N} \times \mathcal{N} \times \mathcal{T}$ with $0 \leq t_1 \leq t_2 \leq ... \leq t_{|\mathcal{G}|}$, where $u_i, v_i \in \mathcal{N}$ are the source and destination node of the $i$-th interaction happening at $t_i \in \mathcal{T}$, respectively.*

Inspired by Huang et al. (2025b), we then define TG link forecasting as a QA task, making it naturally adaptable to LLMs. We discuss the advantages of this formulation over the traditional one in App. E.

**Definition 2 (TG Link Forecasting with LLMs)** *Assume a TG $\mathcal{G} \subseteq \mathcal{N} \times \mathcal{N} \times \mathcal{T}$ containing all ground-truth interactions, and let $f(\cdot)$ denote the inference process of an LLM. Given a prediction query $q = (u_q, ?, t_q)$ with source node $u_q \in \mathcal{N}$ and timestamp $t_q \in \mathcal{T}$, together with its history $\mathcal{H}_{t_q} = \{(u_i, v_i, t_i) \mid t_i < t_q, (u_i, v_i, t_i) \in \mathcal{G}\}$, TG link forecasting requires the model to produce a text-based answer $A$ specifying the ground truth missing node(s) $v_q \subseteq \mathcal{N}$ as the predicted missing destination(s). The answer is obtained by $A = f(\psi(\mathcal{H}_{t_q}, q))$, where $\psi(\cdot, \cdot)$ is a function that converts $\mathcal{H}_{t_q}$ and $q$ into a prompt consisting of historical graph context and a natural language question asking about the missing destination node(s).*

## 3 REAL-TG

The left part of Fig. 1 illustrates our ReaL-TG framework. Given a query $q = (u_q, ?, t_q)$ and its history $\mathcal{H}_{t_q}$ before query timestamp $t_q$, we first apply the Temporal Context Graph Selection (T-CGS) algorithm to construct a subgraph $\mathcal{G}_c$ that is most relevant to $q$ based on $\mathcal{H}_{t_q}$. $\mathcal{G}_c$ serves as the graph context from which the LLM extracts information to make predictions. We then verbalize all links in $\mathcal{G}_c$ and combine them with a natural language question derived from $q$ into a prompt template, denoted as $\mathcal{Q}$. The prompt $\mathcal{Q}$ is fed into an LLM for inference, from which we extract the prediction

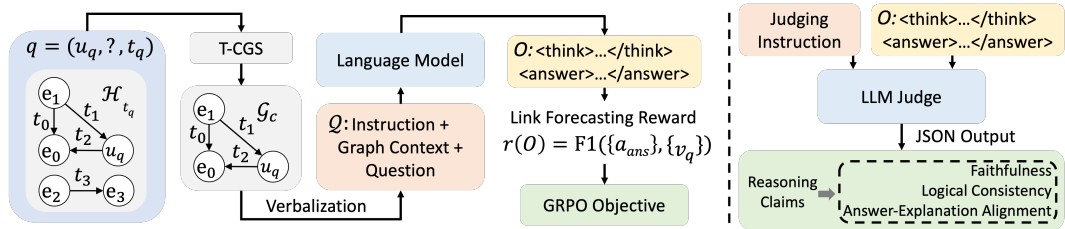

Figure 1: Left: The ReaL-TG framework, which enables RL fine-tuning of LLMs to improve TG forecasting (see Sec. 3). Right: The proposed LLM-as-a-Judge system, which provides a comprehensive evaluation of LLM reasoning quality in TG link forecasting (see Sec. 4, paragraph Reasoning Trace Evaluation).

answer. We compute a link forecasting reward for each prediction with a customized reward function, and through RL the model self-explores reasoning patterns to improve forecasting in TGs.

**Temporal Context Graph Selection.** We input graph context as text into the LLM to ensure explainability, since we require the output reasoning trace to explicitly justify predictions in natural language (see App. F for further discussion on why we represent graph context as text). We aim to include as much relevant graph information as possible while excluding redundant details that do not contribute to prediction. To this end, we propose T-CGS, an algorithm that selects a temporal context graph for each query $q = (u_q, ?, t_q)$. Inspired by Li et al. (2023), we construct $\mathcal{G}_c$ centered around a temporal query node $(u_q, t_q)$. Starting from $(u_q, t_q)$, we perform an $\alpha$-temporal random walk, where at each step the walk terminates at the current temporal node $(e, t)$ with probability $\alpha \in (0, 1)$, and with probability $1 - \alpha$ it continues to a node in the historical temporal neighborhood $Nei_{(e,t)} = (e', t') \mid (e, e', t')$ or $(e', e, t') \in \mathcal{H}_t, t' < t$ of $(e, t)$. If the walk continues, the transition probability from $(e, t)$ to each $(e', t') \in Nei_{(e,t)}$ is given by $P_{(e,t)}(e', t') = \beta^{|\{(e'',t'')|(e'',t'') \in Nei_{(e,t)}, t'' \geq t'\}|} / \sum_{z=1}^{|Nei_{(e,t)}|} \beta^z$, where $\beta \in (0, 1)$ is a decay factor. The intuition behind it is to assign higher transition probabilities to temporal neighbors that are closer in time to the current node $(e, t)$, since recent interactions are generally more influential in information propagation on TGs, as shown in prior works (Liu et al., 2022b; Ding et al., 2022; Li et al., 2023). Based on this setting, we compute the probability of an $\alpha$-temporal random walk starting from the query node $(u_q, t_q)$ and terminating at one of its $k$-hop historical neighbors. We then rank all visited temporal nodes by their termination probabilities and select the top-ranked nodes $\mathcal{N}_q$ as the most relevant for answering query $q$. To construct the context graph $\mathcal{G}_c$, we retrieve all links in the ground-truth graph that involve nodes in $\mathcal{N}q$ and collect them into $\mathcal{G}_c$.

We provide an example in Fig. 2 to show how T-CGS constructs a context graph. Assume we set $\alpha = 0.3$, $\beta = 0.6$ and select only the top-1 temporal node to form $\mathcal{N}q$. For the query node $(u_q, t_q)$, it has two 1-hop temporal neighbors $(e_1, t_1)$ and $(e_2, t_2)$, one 2-hop neighbor $(e_3, t_3)$, and one 3-hop neighbor $(e_2, t_2)$ (this node is both a 1-hop and a 3-hop neighbor), with the temporal order $t_q > t_1 > t_3 > t_2$. The termination probability of $(e_1, t_1)$ is $(1-\alpha)\alpha\beta^2/(\beta+\beta^2) \approx 0.079$, since the random walk first proceeds one step with probability $1 - \alpha$ and then terminates with probability $\alpha$. Similarly, the termination probability of $(e_3, t_3)$ is $(1-\alpha)^2\alpha\beta^2/(\beta+\beta^2) \approx 0.055$. For $(e_2, t_2)$, the termination probability is $(1-\alpha)^3\alpha\beta^2/(\beta+\beta^2) + (1-\alpha)\alpha\beta/(\beta+\beta^2) \approx 0.131$, as it can be reached through two distinct paths. To this end, we have $\mathcal{N}q = \{(e_2, t_2)\}$, and the context graph consists of all the links associated with it, i.e., $\{(u_q, e_2, t_2), (e_3, e_2, t_2)\}$. In practice, we set $|\mathcal{N}q|$ to 100 and limit the random walk to at most 2 steps, yielding a $\mathcal{G}_c$ that contains temporal neighbors of $(u_q, t_q)$ up to 3 hops away. See App. G for more details including the value selection of $\alpha$ and $\beta$.

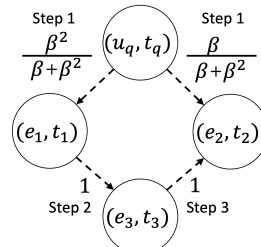

Figure 2: Example of context graph selection.

**Prompt Construction.** Given $\mathcal{G}_c$ and query $q$, we construct the prompt $\mathcal{Q}$ shown in Fig. 3, which embeds the graph context and instructs the LLM to produce both predictions and explanatory

reasoning traces. To facilitate extraction, we require the reasoning to be enclosed within <think> </think> tags and the final predictions within <answer> </answer> tags.

```
<|system|>
You are a temporal graph learning expert.
<|user|>
You will be asked to predict the next interaction (i.e. `Query Destination Node`) given the `Query Source Node` and `Query Timestamp`.
You will also be given a number of historical interactions extracted from a temporal subgraph, where each of them is represented as a tuple of (`Source Node`, `Destination Node`, `Timestamp`). Use this information to predict the most likely `Query Destination Node`s for `Query Source Node` at `Query Timestamp`.
You will only receive information available before `Query Timestamp`. No information at or after this timestamp will be provided. The user instruction is correct and contains no mistakes or typos.
INSTRUCTIONS:
1. You must FIRST think about the reasoning process as an internal monologue and then provide the final answer.
2. The reasoning process MUST BE enclosed within <think> </think> tags.
3. The final answer MUST BE put within <answer> </answer> tags.
4. If the answer contains multiple `Query Destination Node`s, please provide all of them and put them in a list in sorted order, e.g., <answer>[0, 1, 2]</answer>, otherwise, please show the answer as a list with only one element, e.g., <answer>[0]</answer>.

Question:
Given the following historical interactions:
{Links in 𝒢ₑ}
Could you list all plausible `Query Destination Node`s for `Query Source Node` {uq} at `Query Timestamp` {tq}?
```

Figure 3: Prompt template for LLM to do TG link forecasting in ReaL-TG.

**Training Data Collection.** We collect 1,000 link forecasting queries from 4 TGB datasets: `tgbl-wiki`, `tgbl-subreddit`, `tgbl-coin`, and `tgbl-flight` to construct the training data. Since each query $(u_q, ?, t_q)$ may have multiple ground-truth nodes as answers, the total number of involved links is larger than 1,000. Specifically, we sample 225 queries each from `tgbl-wiki` and `tgbl-subreddit`, and 275 queries each from `tgbl-coin` and `tgbl-flight`. The latter two datasets are empirically shown to be more challenging in the original TGB benchmark (Huang et al., 2023), so we allocate more training examples to them. For all datasets, queries are sampled in reverse chronological order from the last training timestamp until the desired size is reached, ensuring richer histories for constructing temporal context graphs. We skip queries where (i) the T-CGS–selected temporal context graph does not contain all ground-truth answers or (ii) the temporal context graph exceeds 600 links. This avoids cases where the LLM cannot observe the answer within its prompt, making fine-tuning meaningless, or where the temporal context graph is so large that it consumes most of the context window, leaving limited space for reasoning. Finally, for each query we construct a $\mathcal{Q}$ prompt and pair it with its ground-truth missing nodes $\{v_q\}$ to form a training example.

**Fine-tuning LLMs with RL.** We use GRPO with a customized reward to fine-tune models. For each query $(u_q, ?, t_q)$ with a set of ground-truth missing nodes $\{v_q\}$, the LLM aims to predict as many ground-truths as possible without introducing spurious nodes. To achieve this, we design a reward function based on the F1 score, balancing precision (whether all predicted nodes are correct) and recall (whether all ground-truth nodes are retrieved). Specifically, let the contents between <answer> </answer> tags in the LLM output $O$ be denoted as $A_{<ans>}$. We parse $A_{<ans>}$ into a set $A = \{a_{<ans>}\}$ of predicted nodes and compute a *link forecasting reward* as

$$r(O) = \text{F1}(\{a_{<ans>}\}, \{v_q\}). \tag{1}$$

This reward depends solely on model outputs, encouraging LLMs to discover transferable reasoning patterns across graphs without constraining their reasoning traces. Moreover, it is non-parametric, requiring no additional cost for training a separate reward model. Given the reward, we update model parameters by maximizing the GRPO objective

$$\mathcal{J}_{\text{GRPO}}(\theta) = \mathbb{E}_{\mathcal{Q} \sim P(\mathcal{Q}), \{O_i\}_{i=1}^g \sim \pi_{\theta_{\text{old}}}(O|\mathcal{Q})} \frac{1}{g} \sum_{i=1}^{g} \frac{1}{|O_i|} \sum_{j=1}^{|O_i|}$$
$$\left( \min \left( \frac{\pi_\theta(O_{i,j}|\mathcal{Q}, O_{i,<j})}{\pi_{\theta_{\text{old}}}(O_{i,j}|\mathcal{Q}, O_{i,<j})} Adv_{i,j}, \text{clip}\left( \frac{\pi_\theta(O_{i,j}|\mathcal{Q}, O_{i,<j})}{\pi_{\theta_{\text{old}}}(O_{i,j}|\mathcal{Q}, O_{i,<j})}, 1-\epsilon, 1+\epsilon \right) Adv_{i,j} \right) - \gamma D_{KL} \left( \pi_\theta \| \pi_{\text{ref}} \right) \right), \tag{2}$$

where $P(\mathcal{Q})$ is the prompt sampling distribution. $\pi_\theta$ and $\pi_{\theta_{\text{old}}}$ denote the current and old policy models[1], respectively. $\epsilon$ is a constant that clips the objective to prevent the policy from changing too

---

[1]In RL, we treat the LLM as a policy model, with the old policy model being the checkpoint before the current update.

drastically in a single update step. $\gamma$ is a weighting factor for the KL-divergence $D_{KL}$ between $\pi_\theta$ and the pre-trained reference model $\pi_{\text{ref}}$, ensuring the fine-tuned model does not diverge excessively from the original base model. For each prompt $\mathcal{Q}$, $g$ rollouts $\{O_i\}_{i=1}^g$ are sampled, each being a full response, and the objective averages over all $|O_i|$ tokens per rollout. $Adv_{i,j}$ denotes the advantage of the $j$-th token in the $i$-th rollout relative to the group of $g$ rollouts, and is defined as $Adv_{i,j} = (r(O_i) - \mu(\{r(O_i)\}_{i=1}^g))/\sigma(\{r(O_i)\}_{i=1}^g)$ where $\mu(\cdot)$ and $\sigma(\cdot)$ denotes mean and standard deviation, respectively. We refer readers to Shao et al. (2024) for more details of GRPO.

## 4 EVALUATION PROTOCOL

We propose a new protocol to evaluate LLMs on TG link forecasting.

**Prediction Label Evaluation.** We first follow Huang et al. (2023) to evaluate models with Mean Reciprocal Rank (MRR). Assume we have $M$ evaluation examples, each consisting of a prompt $Q_m$, a query $(u_{q_m}, ?, t_{q_m})$, and a ground-truth set $\eta_m^{\text{gt}} = \{v_{q_m}\}$. The corresponding prediction set is $\eta_m^{\text{pred}} = \{v'_{q_m}\}$, which contains all nodes the LLM predicts as belonging to $\eta_m^{\text{gt}}$. We compute MRR as follows

$$\text{MRR} = \frac{1}{\sum_{m=1}^M \eta_m^{\text{gt}}} \sum_{m=1}^M \sum_{s=1}^{\eta_m^{\text{gt}}} \frac{1}{\text{rank}_m^s}. \tag{3}$$

$\text{rank}_m^s$ denotes the rank of the $s$-th node in $\eta_m^{\text{gt}}$. The ranking is computed as follows. We first assign a score of 0 to all nodes in the dataset, and then set the score to 1 for nodes included in $\eta_m^{\text{pred}}$. Following prior works (Han et al., 2021; Gastinger et al., 2024), we use filtered MRR, where the influence of other correctly predicted nodes is excluded by resetting their scores to 0 when evaluating a given node. Finally, for each node we compute the mean of its optimistic rank (treating equally scored nodes as ranked lower) and pessimistic rank (treating them as ranked higher), which gives $\text{rank}_m^s$. Although MRR is a widely used and robust metric for evaluating link prediction, it does not capture the risk of *over-generation* in LLMs when the task is framed as QA-style generation. During reasoning, LLMs often predict all nodes they believe belong to $\eta_m^{\text{gt}}$, sometimes accompanied by supporting reasoning. While not always undesirable, this behavior can be problematic when accurate link forecasting is required. To better capture the over-generation phenomenon, we introduce penalized MRR (pMRR), which follows Eq. 3 but slightly modifies the computation of $\text{rank}_m^s$. Specifically, for all nodes in $\eta_m^{\text{pred}} \setminus \eta_m^{\text{gt}}$, we assign a score of 1.1 (can be any number $> 1$) instead of 1. This ensures that incorrectly predicted nodes are ranked above correctly predicted ones, thereby penalizing over-generation. The more such nodes appear, the stronger the penalty, resulting in a lower pMRR.

**Reasoning Trace Evaluation.** LLMs naturally benefit from their text generation ability, making them well-suited for explainable link forecasting. However, no prior work has systematically evaluated their reasoning traces, i.e., how prediction labels are derived. Such evaluation is crucial because a trustworthy forecaster should not only produce accurate predictions but also provide reasonable justifications. Moreover, predictions outside the ground-truth are not always undesirable if they are supported by strong reasoning. In real-world forecasting, ground-truth labels are unavailable before *events actually occur*, unlike in experimental setups where metrics such as MRR can be computed. This makes the evaluation of an LLM forecaster's reasoning quality even more important. The most reliable way to assess LLM reasoning is to do human evaluation, however, it is not scalable. Motivated by the recent success of LLM-as-a-Judge (Zheng et al., 2023), we adopt this approach for quicker and more scalable assessment, focusing on three criteria: faithfulness, logical consistency, and answer–explanation alignment.

- For faithfulness, we evaluate whether the LLM's reasoning is supported by the input context graph $\mathcal{G}_c$. The Judge first splits a reasoning trace into a series of atomic claims, each describing some aspect of the graph context. It then determines the proportion of claims that are faithful to $\mathcal{G}_c$, i.e., contain no factual errors in describing it. This proportion is defined as the faithfulness score $\delta_f$.
- For logical consistency, we assess whether the reasoning follows a coherent and valid chain. Here, the Judge disregards faithfulness and focuses solely on whether the LLM's reasoning proceeds in a logically sound manner without self-contradiction. The Judge assigns a

score from $\{0, 1, 2\}$, with higher values indicating better consistency. This score is then normalized to $[0, 1]$ and defined as the consistency score $\delta_{lc}$.

- For answer–explanation alignment, we assess whether the predicted answers are justified by the model's own reasoning. A predicted node is considered justified if (i) the reasoning trace contains explicit supporting claims for it, and (ii) those claims are judged as faithful in the faithfulness evaluation. We define the alignment score $\delta_a$ as the proportion of predicted nodes that are well-justified.

From another perspective, these three scores can also be viewed as capturing the impact of different types of hallucinations in LLM reasoning. $\delta_f$ targets factual hallucinations, where the model introduces hallucinated claims about the context graph. $\delta_{lc}$ addresses logical inconsistency hallucinations, where the model produces contradictory or incoherent logic chains. $\delta_a$ reflects justification hallucinations, where predictions are made without being grounded in faithful reasoning. By jointly evaluating these dimensions, our system provides a more comprehensive assessment of LLMs' reasoning quality in explainable link forecasting. We use GPT-4.1 mini as a Judge throughout the experiments. See Fig. 4 for the complete prompt, i.e., instruction, for Judge. See the right part of Fig. 1 for an illustration of the system. We compute the aggregated scores $\bar{\delta}_f$, $\bar{\delta}_{lc}$, and $\bar{\delta}_a$ by averaging over all evaluation examples, providing an overall measure of reasoning quality.

## 5 EXPERIMENTS

We fine-tune a Qwen3-4B with ReaL-TG and name our trained model ReaL-TG-4B. We compare it with several baselines on both seen and unseen graphs using our proposed evaluation protocol. We first report comparative results on prediction accuracy and reasoning quality, and a performance comparison between ReaL-TG-4B and traditional TG link forecasting methods (Sec.5.1), followed by further analysis (Sec.5.2) covering: (i) the influence of base model size on ReaL-TG; (ii) human evaluation of reasoning traces from ReaL-TG-4B; and (iii) human evaluation of our LLM-as-a-Judge system. In addition, we include in App.J a qualitative analysis with two case studies demonstrating how RL improves LLM-based link forecasting.

**Experimental Setup.** We collect evaluation data from the test sets of 4 TGB datasets used during training (`tgbl-wiki`, `tgbl-subreddit`, `tgbl-coin`, `tgbl-flight`) and from the test sets of 2 unseen TGB datasets (`tgbl-uci`, `tgbl-enron`) to assess models' transferability to unseen graphs To control evaluation cost, we curate a moderately sized dataset specifically for assessing LLMs in TG link forecasting. We first select the last 1,000 queries from each of the 6 TGB datasets in reverse chronological order, ensuring that test data are accompanied by abundant historical information. For each query, we then extract the temporal context graph using T-CGS. Finally, we filter out queries following the same principles adopted in query skipping when we construct training data and get in total 4,246 evaluation data. The filtering procedure is applied consistently across all datasets, ensuring a fair evaluation that does not introduce bias in comparing different LLMs' capabilities. For baselines, we evaluate several frontier models, including non-reasoning models (Gemma 3 4B/12B, Llama 3.3 70B) and reasoning models (Qwen3-0.6B/4B/8B, GPT5-mini). All models are tested with the same prompts for fair comparison. For non-reasoning models, we use greedy decoding, while reasoning models are run with their default configurations. See App. D for further implementation details.

Table 1: Evaluation data statistics. All data are taken from TGB (Huang et al., 2023) and thus we omit the prefix in dataset names. Inv. means involved, and T means timestamps. Note that we do not reassign node or timestamp IDs; instead, we directly use the anonymized IDs provided in TGB.

| Dataset | # Inv. Nodes | # Queries | # Inv. Links | # Inv. T |
|---|---|---|---|---|
| wiki | 2,844 | 914 | 914 | 17,419 |
| subreddit | 8,097 | 888 | 888 | 44,716 |
| coin | 9,194 | 457 | 482 | 19,792 |
| flight | 5,449 | 488 | 952 | 387 |
| uci | 1,227 | 660 | 660 | 8,738 |
| enron | 296 | 839 | 1,283 | 3,802 |

### 5.1 COMPARATIVE STUDY

**Comparison across Language Models: Prediction Accuracy.** We report the results of MRR and pMRR in Table 2. Our main findings are as follows: (i) within the same model family (e.g.,

Table 2: Comparison across language models: prediction accuracy. The top two results are highlighted by **first** and second.

| | Seen | | | | | | | | Unseen | | | | Combined | |
|---|---|---|---|---|---|---|---|---|---|---|---|---|---|---|
| Dataset | wiki | | subreddit | | coin | | flight | | uci | | enron | | Overall | |
| Model | MRR | pMRR | MRR | pMRR | MRR | pMRR | MRR | pMRR | MRR | pMRR | MRR | pMRR | MRR | pMRR |
| Qwen3-0.6B | 0.338 | 0.331 | 0.245 | 0.238 | 0.111 | 0.107 | 0.121 | 0.111 | 0.114 | 0.108 | 0.089 | 0.084 | 0.171 | 0.164 |
| Qwen3-4B | 0.721 | 0.682 | 0.678 | 0.639 | 0.368 | 0.333 | 0.090 | 0.072 | 0.300 | 0.239 | 0.174 | 0.137 | 0.375 | 0.339 |
| Qwen-8B | 0.763 | 0.721 | 0.731 | 0.688 | 0.380 | 0.343 | 0.109 | 0.087 | 0.364 | 0.293 | 0.300 | 0.243 | 0.436 | 0.391 |
| Gemma 3 4B | 0.698 | 0.673 | 0.686 | 0.650 | 0.290 | 0.235 | 0.159 | 0.121 | 0.328 | 0.268 | 0.274 | 0.223 | 0.407 | 0.364 |
| Gemma 3 12B | 0.782 | 0.738 | 0.718 | 0.671 | 0.376 | 0.302 | 0.315 | **0.249** | 0.390 | 0.298 | 0.469 | 0.381 | 0.520 | 0.452 |
| GPT-5 mini | 0.714 | 0.630 | 0.674 | 0.596 | 0.288 | 0.201 | 0.286 | 0.180 | 0.355 | 0.266 | 0.333 | 0.215 | 0.456 | 0.351 |
| Llama3.3-70B | 0.759 | 0.687 | 0.716 | 0.644 | 0.372 | 0.257 | **0.323** | 0.245 | 0.422 | 0.347 | 0.441 | 0.328 | 0.521 | 0.423 |
| ReaL-TG 4B | **0.824** | **0.792** | **0.765** | **0.726** | **0.431** | **0.401** | 0.198 | 0.175 | **0.607** | **0.523** | **0.492** | **0.435** | **0.552** | **0.508** |

Qwen3-0.6B/4B/8B), larger model size generally leads to better performance on TG link forecasting; (ii) larger LLMs tend to predict more nodes as answers (with larger difference between MRR and pMRR), likely because their stronger capacity allows them to consider more candidate predictions, although this behavior is not always beneficial for link forecasting; (iii) ReaL-TG-4B outperforms all baselines, including GPT-5 mini and Llama 3.3 70B, across nearly all datasets on both seen and unseen graphs, demonstrating the effectiveness of the ReaL-TG framework. Although ReaL-TG-4B trails some baselines on `tgbl-flight`, we attribute this to the limitations of its base model Qwen3-4B on this dataset; (iv) ReaL-TG-4B achieves substantial gains over its base model, confirming the effectiveness of our RL-based training framework.

**Comparison across Language Models: Reasoning Quality.** We report the reasoning evaluation results in Table 3. The comparison includes Qwen3-4B/8B, the Gemma 3 family, and Llama 3.3-70B. We exclude GPT-5 mini for two reasons: (i) our Judge is GPT-4.1 mini, which may introduce family-bias (Spiliopoulou et al., 2025), i.e., assigning higher judgment scores to other OpenAI models; and (ii) the GPT-5 series restricts access to full reasoning traces, providing only a summary of its reasoning, which prevents accurate evaluation of its actual reasoning behavior. We summarize our key findings as follows: (i) within the same model family, larger models are more robust to hallucinations and achieve higher reasoning quality, suggesting a correlation between prediction accuracy and reasoning quality; (ii) ReaL-TG-4B demonstrates

Table 3: Results on the quality of reasoning traces.

| Model | $\bar{\delta_f}$ | $\bar{\delta_{lc}}$ | $\bar{\delta_a}$ |
|---|---|---|---|
| Qwen3-4B | 0.683 | 0.700 | 0.653 |
| Qwen3-8B | 0.792 | 0.808 | 0.770 |
| Gemma 3 4B | 0.595 | 0.666 | 0.558 |
| Gemma 3 12B | 0.867 | 0.928 | 0.771 |
| Llama 3.3 70B | 0.878 | 0.950 | 0.820 |
| ReaL-TG-4B | 0.885 | 0.880 | 0.732 |

substantial improvements over its base model Qwen3-4B in reasoning quality, validating the effectiveness of RL fine-tuning and showing that the ReaL-TG framework enables LLMs to discover meaningful reasoning patterns useful for TG link forecasting; (iii) despite these gains, ReaL-TG-4B lags behind larger models in logical consistency and answer–explanation alignment. We attribute this to the natural advantage of larger models in producing more robust reasoning traces, particularly in providing consistent logic and sufficient supporting evidence for predictions. This indicates that applying ReaL-TG to larger base models would be a promising direction in the future.

**ReaL-TG-4B vs. Traditional TG Link Forecasting Methods.** Table 4 reports results of 3 strong TGNNs: TGN (Rossi et al., 2020), DyGFormer (Yu et al., 2023) and TNCN (Zhang et al., 2024a), together with the widely used EdgeBank baseline (Poursafaei et al., 2022). We train TGNNs separately on the original training set of each involved dataset on TGB with their default implementation settings and evaluate all models using MRR. TGNNs formulate TG link forecasting as a binary classification task, where models are trained to decide whether a potential link exists, which makes ranking metrics computationally expensive since obtaining a rank requires a forward pass over every node in the node set (see App. E for details). Besides, it is impossible to evaluate binary classification-based TGNNs with pMRR because they do not return node IDs directly as answers. To avoid excessive cost, we control the budget for evaluation with a timeout constraint of 24 hours. Note that for ReaL-TG-4B, `tgbl-uci` and `tgbl-enron` are treated as unseen graphs, whereas for TGNNs, they are trained exclusively on these datasets and are therefore considered seen

Table 4: MRR comparison among ReaL-TG-4B and traditional TG link forecasting methods.

| Dataset | wiki | subreddit | coin | flight | uci | enron |
|---|---|---|---|---|---|---|
| EdgeBank | 0.425 | 0.271 | 0.153 | 0.179 | 0.202 | 0.129 |
| TGN | 0.464 | 0.698 | Timeout | Timeout | 0.050 | 0.281 |
| DyGFormer | **0.847** | 0.659 | Timeout | Timeout | 0.011 | 0.341 |
| TNCN | 0.732 | 0.739 | Timeout | Timeout | 0.049 | 0.263 |
| ReaL-TG 4B | 0.824 | **0.765** | **0.431** | **0.198** | **0.607** | **0.492** |

graphs. Our results show that the fine-tuned model outperforms strong traditional methods while providing explicit reasoning to justify its predictions, demonstrating strong potential. Moreover, by formulating TG link forecasting as QA, our framework enables low-cost prediction in real-world applications and eliminates the need to train a model from scratch for new TGs.

## 5.2 FURTHER ANALYSIS

**Influence of Base Model Size on ReaL-TG.** To verify our assumption about the influence of base model size, we also train a separate model, ReaL-TG-0.6B, based on Qwen3-0.6B. We evaluate its reasoning traces with our LLM-based Judge and compare them against Qwen3-4B and ReaL-TG-4B in Table 5. We find that training from a much smaller base model results in significantly worse reasoning quality: even with our RL framework, a 0.6B model is outperformed by a 4B model substantially. Moreover, we observe a notable case of reward hacking (Skalse et al., 2022): in many reasoning traces, the fine-tuned ReaL-TG-0.6B justifies its predictions by claiming "$(u_q, v_q, t_q)$ *has already been seen in the provided graph context*", which is impossible in a forecasting task. This indicates that the model attempts to maximize the outcome-based reward by

Table 5: Results on the quality of reasoning traces compared with ReaL-TG-0.6B.

| Model | $\bar{\delta}_f$ | $\bar{\delta}_{lc}$ | $\bar{\delta}_a$ |
|---|---|---|---|
| ReaL-TG-0.6B | 0.702 | 0.710 | 0.674 |
| Qwen3-4B | 0.683 | 0.700 | 0.653 |
| ReaL-TG-4B | 0.885 | 0.880 | 0.732 |

guessing correct answers while providing a shallow thinking strategy. One major reason is due to the limited reasoning capacity of a tiny model. During RL training, the fine-tuned model must generate full responses (rollouts) based on its own reasoning, following a trial-and-error process guided by the achieved reward. If the base model is too weak, it cannot effectively self-explore more advanced or reasonable reasoning strategies for TG link forecasting. Our results confirm that using a larger base model enables much stronger fine-tuned performance. Nonetheless, we also observe that after fine-tuning with ReaL-TG, the 0.6B model reaches reasoning quality comparable to Qwen3-4B, still highlighting the effectiveness of our RL framework.

**Human Evaluation on the Quality of Reasoning Traces.** We recruit five annotators to evaluate the quality of reasoning traces generated by ReaL-TG-4B. A random sample of 50 data examples is selected, and annotators provide judgment scores for the three criteria following the same instructions given to the LLM-based judge. Averaging their annotations yields high scores of 0.885/0.872/0.839 for $\bar{\delta}_f/\bar{\delta}_{lc}/\bar{\delta}_a$ (maximum score 1), which closely align with the judge's scores of 0.909/0.890/0.787 (annotation variances are 0.001/0.004/0.001). This strong correlation not only validates our LLM-as-a-Judge system but also demonstrates the substantial reasoning capability gained through fine-tuning with ReaL-TG. Further annotation details are provided in App. I.

**Human Evaluation on the Quality of the LLM-as-a-Judge System.** To directly assess the reliability of our LLM-based judging system, we use the same 50 samples and collect both the responses generated by ReaL-TG-4B and the corresponding judgments from the system. We ask the same five human annotators to evaluate the quality of these judgments. For each of the three criteria, annotators assign a score from {0,1,2}, with higher values indicating better judging quality. The resulting average scores are 1.71 for faithfulness, 1.88 for logical consistency, and 1.71 for answer–explanation alignment (maximum 2, and variances are 0.016, 0.013 and 0.014, respectively), demonstrating excellent judgment quality. Due to cost constraints, we employ GPT-4.1 mini as the judge, however, judging quality is strongly tied to the capability of the underlying model (Huang et al., 2025a) and can be enhanced by switching to a more advanced judge, such as Gemini 2.5 Pro.

## 6 CONCLUSION

In summary, we present ReaL-TG, the first RL-based framework that enables LLMs to perform explainable and effective link forecasting on TGs. We further introduce a new evaluation protocol, featuring a new automated ranking metric coupled with a dedicated LLM-as-a-Judge system. Our experiments show that ReaL-TG allows LLMs to self-explore reasoning strategies for TG link forecasting, achieving improvements both in prediction accuracy and in generating well-grounded reasoning traces. We also conduct human evaluation of both the LLM-as-a-Judge system and the fine-tuned model, validating the effectiveness of our framework and evaluation methodology.

ETHICS STATEMENT

Our work applies LLMs to TG link forecasting, and thus inherits the well-known risks associated with LLMs. For instance, LLMs are prone to hallucination, often producing responses that appear plausible but are factually incorrect. While we show that ReaL-TG can mitigate hallucination to some extent, it cannot eliminate it entirely. Therefore, practitioners adopting ReaL-TG should remain aware of these behaviors and exercise caution in fully trusting LLM outputs, especially in safety-critical applications where misuse or overreliance could lead to adverse outcomes in ethics.

REPRODUCIBILITY STATEMENT

We have uploaded our source code and curated QA dataset for training, validation, and test in the Supplementary Material. It also includes detailed instructions for environment setup, training, LLM generation, evaluation, and LLM judging in an enclosed `README.md`, enabling readers to reproduce our experimental results. In addition, we provide details on dataset access from TGB in App.C, as well as implementation details of ReaL-TG-based LLM training, evaluation, LLM judging, and TGNNs in App.D.

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

## A  THE USE OF LARGE LANGUAGE MODELS

We use LLMs to assist paper writing by refining the human-written contents. We further use LLMs to help refine our prompt templates shown in Fig. 3 (Section 3) and 4 (Section 4). LLMs are also used in App. I to refine the human annotation guideline in Fig. 5.

## B  LIMITATIONS

The capabilities of LLMs fine-tuned with ReaL-TG are inherently limited by the input temporal context graph. If key predictive signals lie outside the $k$-hop historical neighborhood considered in T-CGS, ReaL-TG may struggle to identify the correct solution. Similar limitations are observed in many TGNN models, which also rely on temporal neighbor sampling to select the most informative neighbors for aggregation Rossi et al. (2020); Xu et al. (2020). In addition, LLMs are constrained by their context window size, which limits the amount of temporal graph information they can process. For instance, the base model used in our work, Qwen3-4B, has a context window of 32k tokens, making it infeasible to provide entire real-world TGs as input. We also provide a more detailed discussion about this problem in App. F.

## C  DATASET ACCESS

All datasets used in this work is obtained from the Temporal Graph Benchmark Github repository[2]. The TGB package provides download links along with the processed files. Some datasets used in this work was added in recent updates to TGB such as `subreddit`, `uci` and `enron`. The download links for the datasets from TGB are as follows: `tgbl-wiki`[3], `tgbl-subreddit`[4], `tgbl-coin`[5], `tgbl-flight`[6], `tgbl-uci`[7], `tgbl-enron`[8].

## D  IMPLEMENTATION DETAILS

**Training.**  We train ReaL-TG-4B with Qwen3-4B as the base model. We develop ReaL-TG on top of verl (Sheng et al., 2024), a strong framework for post-training on language models. Our training is performed on a compute node with 96 Intel(R) Xeon(R) Platinum 8469C CPU cores and 4 × Nvidia H100 GPU each with 80GB memory. We provide the training hyperparameters in Table 6.

Table 6: Hyperparameter configurations of ReaL-TG-4B during training.

| Model | # Epoch | Batch Size | Mini-Batch Size | Learning Rate | $\gamma$ | Max Response Length | # Rollout ($g$) |
|---|---|---|---|---|---|---|---|
| ReaL-TG-4B | 3 | 32 | 16 | $2e^{-6}$ | 0.001 | 16,384 | 5 |

**Evaluation.**  All evaluations are conducted on the same compute node as used for training. For the Qwen3 family, we generate responses using verl, following their official repositories: Qwen3-0.6B[9], Qwen3-4B[10], and Qwen3-8B[11]. The Gemma 3 family is run via Hugging Face Transformers (Wolf et al., 2019), using their official repositories: Gemma-3-4B-it[12] and Gemma-3-12B-it[13]. We also evaluate Llama-3.3-70B[14] under the same setting. For GPT-5-mini, we use OpenAI's openai-python API. The specific release we use in our experiments is gpt-5-mini-2025-08-07. All reasoning models are executed three times with default hyperparameters, and we report the mean results. Non-reasoning models are run with temperature fixed to 0 for greedy decoding, while all other hyperparameters follow their default configurations.

**Judge Model.**  We employ GPT-4.1-mini for our LLM-as-a-Judge system, implemented via OpenAI's openai-python API. Specifically, we use the gpt-4.1-mini-2025-04-14 release in our experiments. To ensure reproducibility, the model's temperature is set to 0, and outputs are constrained to JSON format for reliable parsing of judgment information.

**TGNN Baselines.**  For training the baseline TGNN models, we use NVIDIA A100 GPUs (80GB memory) paired with 4 CPU nodes (2.65 GHz, 128MB L3 cache), each equipped with 128GB RAM. When the experiments runs more than 24 hours, we consider it to reach timeout to avoid excessive cost. We use the TGB implementation of baselines with their default hyperparameters. Each model is trained on the complete TGB training set and then validated on the TGB validation set when searching for the best checkpoint.

---

[2]https://github.com/shenyangHuang/TGB
[3]https://object-arbutus.cloud.computecanada.ca/tgb/tgbl-wiki-v2.zip
[4]https://object-arbutus.cloud.computecanada.ca/tgb/tgbl-subreddit.zip
[5]https://object-arbutus.cloud.computecanada.ca/tgb/tgbl-coin-v2.zip
[6]https://object-arbutus.cloud.computecanada.ca/tgb/tgbl-flight-v2.zip
[7]https://object-arbutus.cloud.computecanada.ca/tgb/tgbl-uci.zip
[8]https://object-arbutus.cloud.computecanada.ca/tgb/tgbl-enron.zip
[9]https://huggingface.co/Qwen/Qwen3-0.6B
[10]https://huggingface.co/Qwen/Qwen3-4B
[11]https://huggingface.co/Qwen/Qwen3-8B
[12]https://huggingface.co/google/gemma-3-4b-it
[13]https://huggingface.co/google/gemma-3-12b-it
[14]https://huggingface.co/meta-llama/Llama-3.3-70B-Instruct

# E    ADVANTAGES OF QA FORMULATION FOR TG LINK FORECASTING

Previous studies typically formulate TG link forecasting as a binary classification task, where models are trained to determine whether a potential link $(u_q, v_q, t_q)$ exists.

**Definition 3** *Given a TG $\mathcal{G}$, a source node $u_q \in \mathcal{N}$, a destination node $v_q \in \mathcal{N}$, a timestamp $t_q \in \mathcal{T}$, together with the history $\mathcal{H}_{t_q} = \{(u_i, v_i, t_i) \mid t_i < t_q, (u_i, v_i, t_i) \in \mathcal{G}\}$, TG link forecasting aims to predict whether the interaction $(u_q, v_q, t_q)$ exists.*

This makes the computation of ranking metrics such as MRR highly costly. To obtain the rank of a node $e \in \mathcal{N}$, the model must perform a forward pass for every candidate node in $\mathcal{N}$, resulting in a total of $|\mathcal{N}|$ passes that scale with $|\mathcal{N}|$ linearly. In contrast, by formulating TG link forecasting as a QA problem, the model can directly output the predicted nodes in a single forward pass, substantially reducing computational cost for real-world TGs with large $|\mathcal{N}|$. In TGB (Huang et al., 2023), for each existing *positive* link in the evaluation data, Huang et al. sample a set of *negative* links with false destination nodes and compare the model scores assigned to them. Their evaluation does not consider all nodes in $|\mathcal{N}|$. In contrast, in this work, both MRR and pMRR are computed against the entire node set $|\mathcal{N}|$, which ensures evaluation completeness and efficiency.

# F    CAN WE INJECT GRAPH CONTEXT IN OTHER WAYS?

A limitation of our approach of injecting graph context purely as text is that the amount of information is constrained by the LLM's context window. Several works instead compress graphs into low-dimensional representations and jointly fine-tune them with language models (Chai et al., 2023; Chen et al., 2024). While effective for downstream tasks, this strategy faces a key limitation for explainable link forecasting. In principle, one could compress more graph information—including the entire historical graph—into such representations, giving LLMs maximal input coverage. Although this offers an advantage over our text-based method, overly compressed representations make it difficult for LLMs to distinguish relevant information for prediction from redundant details. Furthermore, explainable forecasting requires human-readable reasoning traces that depend directly on the input graph context. If the graph is not provided as text, the LLM must also learn to reconstruct graphs from encoded representations back into natural language during inference, which is possible but would require substantial methodological advances. We regard the problem of optimally providing graph context for LLMs as outside the scope of this work, but an important open direction for future research.

# G    T-CGS DETAILS

**Parameter Setting of $\alpha$ and $\beta$.**    We choose the values of $\alpha$ and $\beta$ to balance the selection of nodes across different historical distances and hop counts from the query node. A larger $\beta$ makes it less likely to select nodes from more distant history, while a larger $\alpha$ reduces the likelihood of selecting nodes from farther hops. We then construct a search grid for $\alpha$ and $\beta$ with candidate values $\{0.1/0.4, 0.3/0.6, 0.5/0.8, 0.7/0.9\}$. For each setting, we construct context graphs on the last 1000 training samples of `tgbl-coin` and collect statistics of the selected nodes. The configuration $0.3/0.6$ yields the best balance, ensuring that the selected nodes are neither overly concentrated in very recent history and first-hop neighbors nor excessively dispersed away from them. Thus, we set $\alpha = 0.3$ and $\beta = 0.6$ for all of our experiments in ReaL-TG.

# H    FULL PROMPTS

# I    HUMAN EVALUATION AND ANNOTATION DETAILS

We recruit 5 human annotators to do evaluation on the quality of our LLM-as-a-Judge system as well as the reasoning traces output by our fine-tuned ReaL-TG-4B. All annotators are either PhD students or Postdoctoral Researchers in Computer Science with at least full professional proficiency in English. All of them consent our usage of their data. The annotation guidelines are provided in Fig. 5.

```
<|system|>
You are a meticulous evaluator for temporal graph QA with explanations.
You will receive: (q) the question, (g) a temporal subgraph as lines of (src, dst, ts) strictly before the query timestamp
and a model response R that contains an explanation inside <think>...</think> and a final answer list inside
<answer>...</answer>
Your job is to output ONLY valid JSON matching the JSON Schema provided in the instructions.
You should follow the evaluation procedure as follows:

---
### Evaluation Procedure

1. Parse response
- Extract the answer list A as the JSON array inside <answer> ... </answer>. If parsing fails, set A=[] and record a note
in alignment.notes.
- Extract the explanation E as the natural-language content inside <think> ... </think>. Judge only what is explicitly
stated in E.

2. Break explanation into atomic claims
- Split E into minimal atomic claims about edges, nodes, times, paths, counts, or membership related to graph.
- Produce a list of claims = [c1, c2, ...]. Use short, verifiable sentences.
- Also return the number of claims as #claims.

3. Faithfulness to g
- For each claim ci, label one of:
  "Supported" (entailed by g),
  "Contradicted" (g states the opposite),
  "Not-in-g" (cannot be verified from g; count as unsupported).
- faithfulness.score = #Supported / max(1, #claims).
- Return all Supported claims. For non-Supported claims, return objects with fields: claim, reason ("Contradicted"|"Not-
in-g"), and pointer (cite/summarize lines in g).

4. Logic Consistency (internal reasoning soundness; independent of g's truth)
- Use 0-2 scale:
  2 Excellent - steps are necessary & sufficient; no contradictions; valid transitions; no major gaps.
  1 Good - slight gap or mild unstated assumption; mostly valid.
  0 Poor/Invalid - The reasoning is unconvincing or fundamentally flawed. It may have significant gaps, make speculative
leaps, contain inconsistencies, or include clear formal fallacies like contradictions or circular reasoning.
- Return a rationale with a concise one-sentence summary.

5. Answer-Explanation Alignment
- An answer a ∈ A is justified iff:
  (1) E explicitly argues for a, and
  (2) those supporting claims are Supported in step 3.
- alignment.score = |justified_answers| / max(1, |A|).
- Return the justified_answers.
- Return the justification_notes that explicitly indicates why the answers are justified. This part will be used to
classify the reasoning patterns of models, so be clear and concise.
- Return the unjustified_answers (in A but not justified).

6. Output
- Return ONLY a JSON object with fields: claims, faithfulness, logic, alignment.
- Do not include any text outside the JSON object.
---

Score three aspects: (1) Faithfulness to g, (2) Logic Consistency, (3) Answer-Explanation Alignment.

IMPORTANT INSTRUCTIONS:
1. Please be VERY CAUTIOUS when you are asked to extract claims and calculate the number of claims.
2. When you are asked to extract claims, DO NOT include any claim making conclusions about the final answer.
3. In many cases, model will correct its previous claims with new claims during reasoning. When you are asked to extract
claims, ALWAYS consider this situation and ONLY include the claims that are not corrected by the model in later steps.
4. When you are asked to evaluate logic consistency, you should evaluate the explanation as a whole reagrdless of the
result of faithfulness.
5. The timestamps with larger numbers are later than the ones with smaller numbers.
6. When judging whether answers are justified or writing justification_notes, remain strictly objective and evaluate only
against the model's own explanation. Consider an answer justified if the explanation explicitly supports it, even if you
personally disagree with the reasoning. DO NOT mark an answer as unjustified simply because you think it should be
justified in another way.

<|user|>
### JSON Schema
Your output must be a single JSON object that validates against this schema:
{schema_json}

### Inputs
- q:
Could you list all plausible `Query Destination Node`s for `Query Source Node` {u_q} at `Query Timestamp` {t_q}?

- g (historical interactions; all timestamps < {ts}):
{Links in G_c}

- Metadata:
  - Query Source Node: {u_q}
  - Query Timestamp: {t_q}
  - Ground-truth answers: {{v_q}}
  - Model's final answer: {{a_<ans>}}

- Model response R:
{O}
```

Figure 4: Prompt template for LLM-as-a-Judge system.

```
                        ReaL—TG Human Annotation Guideline

Data: Please download it and fill it out locally
Background:
You will be given multiple data examples. Each consists of:
— A prompt input into a language model (LM).
— The LM's response.
— The judgement of LM's response produced by an automated judging system according to the
  following  procedure:

———
### Evaluation Procedure...
———

Your need to perform two tasks:
Task 1: Judge Evaluation
Evaluate whether the judging system's scores (faithfulness, logical consistency, alignment) are reasonable. For each score,
assign one label:
0 — Big mistake / false
— The score is not supported by correct reasoning if reasoning trace is given.
— The score is incorrectly assigned since it does not match the logic behind the judging system's reasoning.

1 — Largely correct (minor fault)
— The score largely reflects the true quality of the LM response.
— If available, the explanation for the score is mostly accurate but may contain small imprecisions, minor omissions, or
slightly unclear reasoning.

2 — Completely correct
— The score and reasoning are fully correct and accurately reflect the LM's response quality.

Task 2: Human Judge
Re—evaluate each example yourself using the same procedure. You only need to output the three scores (faithfulness, logical
consistency, alignment). If the judging system's score is completely correct, you may directly copy it without re—
evaluating.
```

Figure 5: Human Annotation guideline. The detailed evaluation procedure is taken from the prompt template for the LLM-based judging system in Fig. 4.

## J QUALITATIVE ANALYSIS: HOW DOES RL HELP?

From Table 2 and 3, we observe consistent improvements of the ReaL-TG-trained model over its base model. To illustrate what the model has learned through RL that leads to these gains, we provide a qualitative analysis based on two case studies, comparing ReaL-TG-4B and Qwen3-4B. In Case 1 (Fig. 6 and 7), we observe that after RL, the model no longer exhausts the context window by repeating the same content. Instead, it predicts the most plausible destination node by leveraging interaction recency. In Case 2 (Fig. 6 and 8), we observe that after RL, the model is less prone to getting stuck in iterative self-reflection and demonstrates greater confidence and effectiveness in applying reasoning strategies to support its predictions. To summarize, exploration during RL, in which an LLM tries different strategies for forecasting links depending on the observed graph context, is essential for improving both prediction accuracy and the quality of reasoning traces. Although base models already show strong abilities in producing plausible reasoning, they still need to learn how to adjust their reasoning style to the specific context in which it is applied.

## K TRAINING CURVES

We provide two curves, Reward vs. Training Step and Validation F1 Score vs. Training Step in Fig. 9 and 10. Our validation is conducted on 500 examples uniformly sampled from the validation sets of the 4 datasets used for training. We use F1 score as metrics during validation. From Fig. 9, we observe that the reward increases with training steps and eventually reaches a plateau, indicating that training has saturated. From Fig. 10, we observe that the validation trend is consistent with the reward curve, and in our experiments, we select the checkpoint with the best validation performance as the final model for evaluation.

## L QUANTIFICATION OF REWARD HACKING

As mentioned in Sec. 5.2, in many reasoning traces, the fine-tuned ReaL-TG-0.6B justifies its predictions by claiming something like "$(u_q, v_q, t_q)$ *has already been seen in the provided graph context*", which can be interpreted as a type of reward hacking in a forecasting task. To further

**Case1**

$\mathcal{G}_c$:
(3390, 8648, 833529), (3390, 8648, 927657), (4272, 8929, 1027429), (4272, 8929, 1027461), (104, 8648, 1093360), (3390, 8648, 1103097), (3390, 8648, 1103671), (167, 8648, 1266808), (167, 8648, 1266809), (866, 8648, 1278569), (4459, 8648, 1335789), (4459, 8648, 1335874), (3390, 8929, 1344764), (3390, 8648, 1344818), (3390, 8648, 1344868), (4459, 8648, 1353699), (4459, 8648, 1353719), (866, 8648, 1389561), (866, 8648, 1390132), (866, 8648, 1420514), (997, 8929, 1444089), (997, 8929, 1444395), (997, 8929, 1446670), (997, 8929, 1446795), (997, 8929, 1450527), (423, 8648, 1451994), (3390, 8929, 1461814), (3390, 8648, 1463750), (859, 8648, 1504113), (866, 8648, 1517985), (866, 8648, 1518071), (866, 8648, 1518498), (866, 8648, 1519023), (997, 8929, 1522620), (2727, 8648, 1524334), (866, 8648, 1525088), (866, 8648, 1525235), (5522, 8929, 1525556), (2863, 8929, 1533240), (997, 8929, 1534720), (2863, 8929, 1535928), (2863, 8929, 1535943), (4531, 8929, 1536373), (3390, 8929, 1547848), (3390, 8648, 1549002), (233, 8648, 1575061), (4459, 8648, 1590422), (4459, 8648, 1593828), (611, 8648, 1596720), (5937, 8648, 1606417), (5937, 8648, 1606438), (5937, 8648, 1606461), (5938, 8648, 1607964), (5938, 8648, 1608194), (3390, 8648, 1620262), (997, 8929, 1620278), (997, 8929, 1620574), (997, 8929, 1620852), (997, 8929, 1621381), (997, 8929, 1622753), (997, 8929, 1622892), (5522, 8929, 1624366), (5522, 8929, 1624414), (997, 8929, 1624661), (997, 8929, 1628002), (997, 8929, 1657475), (3390, 8929, 1691346), (997, 8929, 1695077), (997, 8929, 1695521), (3390, 8929, 1696857), (6942, 8929, 2061590), (997, 8929, 2062009), (997, 8929, 2133359), (997, 8929, 2133419), (5522, 8929, 2218607), (7458, 8929, 2262998), (7458, 8929, 2264131), (7458, 8929, 2264356), (7458, 8929, 2264753), (7458, 8929, 2265033), (997, 8929, 2283892), (997, 8929, 2283988), (3390, 8929, 2289548), (8173, 8929, 2646640), (8173, 8929, 2646702), (997, 8929, 2648320), (5522, 8929, 2656128), (997, 8929, 2656490), (8192, 8929, 2659851), (8192, 8929, 2659898), (8192, 8929, 2660147), (8192, 8929, 2660185), (17, 8929, 2660187), (997, 8929, 2663130), (997, 8929, 2663161), (8199, 8929, 2664402), (3390, 8929, 2677842)
q: (3390, ?, 2677935)
$\{v_q\}$: {8929}

**Case 2**

$\mathcal{G}_c$:
(574, 8552, 1419500), (574, 8552, 1419845), (1601, 8552, 1420897), (3458, 8552, 1432139), (5539, 8552, 1448204), (5539, 8552, 1448331), (1726, 8552, 1458033), (5204, 8552, 1502319), (1206, 8552, 1505338), (2466, 8852, 2315899), (221, 9149, 2439895), (7854, 8852, 2460397), (3138, 9149, 2473041), (1206, 9149, 2473942), (499, 9149, 2479422), (1206, 8734, 2481811), (1206, 8852, 2481993), (499, 9149, 2484302), (221, 9149, 2489612), (4096, 8734, 2501385), (5528, 8734, 2501601), (4096, 8734, 2501828), (1942, 8852, 2502029), (1187, 8734, 2508169), (1206, 8734, 2508797), (1206, 8734, 2509084), (1206, 8734, 2509168), (1206, 8734, 2509314), (1206, 8734, 2509471), (221, 9149, 2515672), (221, 9149, 2516310), (221, 9149, 2517110), (221, 9149, 2518569), (7959, 8734, 2522021), (221, 8734, 2526640), (221, 8734, 2528137), (1221, 8734, 2531985), (1221, 8734, 2532364), (1221, 8734, 2532652), (1343, 8734, 2536121), (1369, 8734, 2539035), (1206, 8734, 2539495), (1206, 8734, 2539603), (2466, 8852, 2561406), (2210, 8734, 2564667), (7914, 8734, 2566838), (8035, 8552, 2567081), (2761, 8552, 2575312), (1680, 8734, 2579425), (1206, 9149, 2586472), (1206, 9149, 2586707), (8035, 8552, 2591725), (1680, 8734, 2593063), (1680, 8734, 2593653), (1680, 8734, 2593703), (1680, 8734, 2594042), (1680, 8734, 2594277), (1680, 8734, 2594499), (4554, 8734, 2597331), (2723, 8852, 2603595), (2723, 8852, 2603688), (2723, 8852, 2603764), (2723, 8852, 2603801), (2723, 8852, 2603995), (221, 8734, 2606232), (1343, 8734, 2614416), (4752, 9084, 2626153), (1973, 9084, 2626233), (1973, 9084, 2626264), (1973, 9084, 2626290), (2425, 9084, 2626614), (8135, 9084, 2626712), (480, 9084, 2626802), (480, 9084, 2626960), (480, 9084, 2627053), (480, 9084, 2627129), (1206, 9084, 2627443), (1206, 8734, 2628541), (1206, 8734, 2628595), (4085, 8552, 2644290), (1206, 8552, 2646542), (4638, 8852, 2657228), (4638, 8852, 2657343), (4638, 8852, 2657416), (4638, 8852, 2658765), (4638, 8852, 2659045), (4638, 8852, 2661840), (4638, 8852, 2662118), (1206, 8852, 2670794), (1206, 8852, 2670884)
q: (1206, ?, 2670969)
$\{v_q\}$: {8852}

Figure 6: The verbalized temporal context graph, query, and ground truths in Case 1 and 2. The input prompts to two cases are derived from combining contents in this figure with the prompt template in Fig. 3.

investigate this issue, we conduct an additional evaluation using GPT-4.1 (temperature is set to 0) with the prompt shown in Fig. 11. We evaluated all responses across the full evaluation set.

| ReaL-TG-4B | Qwen3-4B |
|---|---|
| <think> | <think> |
| Okay, let's tackle this problem. The user is asking for the most likely destination node for a query where the source node is 3390 and the timestamp is 2,677,935. **First, I need to look at all the historical interactions involving node 3390 and see if there's a pattern or frequent destination nodes around the query timestamp.** The query timestamp is 2,677,935. Let's check each entry where the source node is 3390... Looking at the list, I see an entry: (3390, 8929, 2677842). The timestamp here is 2,677,842, which is before the query timestamp... So, **this interaction happened recently. The destination node is 8929...** But maybe there are other interactions where source is 3390... For example, earlier entries: like (3390, 8648, 1620262), (3390, 8929, 1691346), (3390, 8929, 2677842). So, three interactions where source is 3390. Let's check their timestamps. The first is at 1,620,262 (before 2.6m), second at 1,691,346 (before 2.6m), third at 2,677,842 (just before query time). **So, the most recent is with 8929 at 2.6m. So, the answer would be 8929.** </think> <answer> [8929] </answer> | Okay, let's tackle this problem. The user wants to predict the next destination node for source node 3390 at timestamp 2677935 based on historical interactions. First, I need to look at all the historical data provided. Looking through the list, I notice that 3390 is the source in several interactions. Let me list those out. The historical entries where the source is 3390 are: (3390, 8648, 833529) (3390, 8648, 927657) (3390, 8929, 1027429)... (3390, 8648, 1344868) **(3390, 8648, 1344868)** **(3390, 8648, 1344868)** **(3390, 8648, 1344868)...** |

Figure 7: Comparison of reasoning traces on the same evaluation data generated by ReaL-TG-4B and Qwen3-4B on Case 1. The **black bold** part indicates the effective reasoning strategy used by ReaL-TG-4B to obtain the correct prediction, whereas the **red bold** part marks the low-quality reasoning produced by Qwen3-4B. Qwen3-4B failed to complete its reasoning and instead filled the context window by repeatedly generating the same link.

We computed the proportion of cases in which this phenomenon appears for ReaL-TG-0.6B and ReaL-TG-4B. The 4B model shows a very low rate of 5.6%, whereas the 0.6B model reaches 67.3%. This substantial gap confirms our conclusion that smaller models are more susceptible to such behavior during RL. We also manually inspected 30 randomly sampled responses from ReaL-TG-0.6B, ReaL-TG-4B, and the base Qwen3-4B. We found that Qwen3-4B never exhibits this behavior, which is expected because baseline models are not RL fine-tuned on our task-specific data and therefore cannot develop this RL-induced pattern. Additionally, the manual inspection results fully align with GPT-4.1's judgments, showing strong agreement between human annotation and our automated evaluation.

Importantly, even when this phenomenon appears, it does not compromise the integrity of our LLM-as-a-Judge evaluation. In practice, statements such as *"$(u_q, v_q, t_q)$ has already been seen"* appear only once or twice in a response and do not dominate the reasoning. Our faithfulness rubric evaluates all atomic claims, and combines their results together into one score, so isolated improper statements carry limited weight. Nonetheless, we believe this analysis is a valuable supplement for understanding how such reward hacking-related behavior emerges under RL.

## M  DOES REAL-TG HURT REASONING IN OTHER DOMAINS?

To study whether ReaL-TG hurts models' reasoning capabilities other than TG forecasting, we evaluate Qwen3-4B and ReaL-TG-4B on GSM8K (Cobbe et al., 2021) and MATH-500 (Hendrycks et al., 2021), two standard benchmarks for evaluating the mathematical reasoning capabilities of LLMs. The metric used for both datasets is accuracy. Both models are evaluated with identical hyper-parameters, including temperature set to 0.6, top-p set to 0.95, and top-k set to 20, as recommended by the official Qwen3 technical report. We set the maximum output length to 30,000 tokens to ensure that long-form reasoning is not truncated before producing a final answer. For evaluation, we used

EvalScope (Team, 2024), a widely used LLM evaluation framework, which provides standardized and reproducible evaluation implementations. MATH-500 includes 5 level of problems, so here we also include level-wise results. The results are presented in the Table 7.

We observe that our fine-tuned model does not sacrifice mathematical reasoning capabilities. In fact, it even shows slight improvements in most settings. We believe the reason is the following. During training, we explicitly impose a KL divergence-based loss between the reference model, which is the original Qwen3, and the fine-tuned model. By inspecting our training logs, we found that at the final training step the KL loss was as small as 0.00025. This indicates that our model has not drifted far from the original Qwen3, and therefore should not behave very differently in domains beyond our training target, namely, temporal graph link forecasting. At the same time, we do not want to overclaim. We are not proving that our training procedure improves general mathematical reasoning because the observed gains are small and not the focus of our method. Our motivation is solely to enhance LLMs for link forecasting, and broader gains are outside our scope. The purpose of this comparison is simply to provide empirical evidence that our fine-tuning does not harm the model's general reasoning ability.

Table 7: Qwen3-4B vs. ReaL-TG-4B on GSM8K and MATH-500.

| Dataset | Subset | # Data Instances | Qwen3-4B | ReaL-TG-4B |
|---------|--------|------------------|----------|------------|
| GSM8K | Overall | 1319 | 0.948 | 0.949 |
| MATH-500 | Level 1 | 43 | 0.954 | 0.977 |
| MATH-500 | Level 2 | 90 | 0.978 | 0.978 |
| MATH-500 | Level 3 | 105 | 0.971 | 0.952 |
| MATH-500 | Level 4 | 128 | 0.953 | 0.961 |
| MATH-500 | Level 5 | 134 | 0.903 | 0.910 |
| MATH-500 | Overall | 500 | 0.948 | 0.950 |

# N    CAN MODEL LEARN TEMPORAL GRAPH REASONING FROM STATIC GRAPHS?

We provide here an additional analysis demonstrating that a dedicated framework for TG reasoning is crucial. Even when LLMs are fine-tuned with RL to improve their reasoning on static graphs, they still fail to effectively learn how to reason over TGs.

G1 (Guo et al., 2025) is a notable concurrent work that demonstrates the effectiveness of RL fine-tuning on static graph reasoning tasks at scale (100k training examples). To show whether such large-scale static graph training is sufficient for TG forecasting, we tested G1-3B[15] and G1-7B[16] on our evaluation set using their default hyperparameters, and compare them with Qwen3-4B and our ReaL-TG-4B. We report the experimental results in Table 8 and 9.

Table 8: Comparison across Qwen3-4B, G1-3B, G1-7B and ReaL-TG-4B: prediction accuracy. The top two results are highlighted by **first** and second.

| Dataset | Seen | | | | | | | | Unseen | | | | Combined | |
|---------|------|------|------|------|------|------|------|------|------|------|------|------|------|------|
| | wiki | | subreddit | | coin | | flight | | uci | | enron | | Overall | |
| Model | MRR | pMRR | MRR | pMRR | MRR | pMRR | MRR | pMRR | MRR | pMRR | MRR | pMRR | MRR | pMRR |
| Qwen3-4B | 0.721 | 0.682 | 0.678 | 0.639 | 0.368 | 0.333 | 0.090 | 0.087 | 0.300 | 0.239 | 0.174 | 0.137 | 0.375 | 0.339 |
| G1-3B | 0.650 | 0.641 | 0.642 | 0.629 | 0.299 | 0.286 | 0.178 | 0.161 | 0.376 | 0.348 | 0.270 | 0.246 | 0.382 | 0.382 |
| G1-7B | 0.794 | 0.782 | **0.786** | **0.770** | 0.383 | 0.331 | 0.193 | 0.151 | 0.485 | 0.445 | 0.464 | 0.398 | 0.523 | 0.484 |
| ReaL-TG-4B | **0.824** | **0.792** | 0.765 | 0.726 | **0.431** | **0.401** | **0.198** | **0.175** | **0.607** | **0.523** | **0.492** | **0.435** | **0.552** | **0.508** |

From these experiments, we observe a consistent pattern: (i) static graph RL helps, but does not solve temporal forecasting. G1-3B improves over Qwen3-4B, showing that RL on graph reasoning data is indeed useful. However, both G1 models still lag behind ReaL-TG-4B on almost all datasets,

---

[15]https://huggingface.co/PKU-ML/G1-3B

[16]https://huggingface.co/PKU-ML/G1-7B

Table 9: Results on the quality of reasoning traces compared across Qwen3-4B, G1-3B, G1-7B and ReaL-TG-4B.

| Model | $\bar{\delta}_f$ | $\bar{\delta}_{lc}$ | $\bar{\delta}_a$ |
|-------|------|------|------|
| Qwen3-4B | 0.683 | 0.700 | 0.653 |
| G1-3B | 0.685 | 0.692 | 0.600 |
| G1-7B | 0.859 | 0.872 | **0.750** |
| ReaL-TG-4B | **0.885** | **0.880** | 0.732 |

especially on `tgbl-uci` and `tgbl-enron`; (ii) even large-scale static graph RL does not transfer to temporal graphs. Despite using around 100 times more training instances, G1-7B still underperforms ReaL-TG-4B, which was trained on only 1k TG training instances. This indicates that TGs introduce reasoning challenges that simply do not arise in static graph settings, and thus cannot be learned from static graph data alone; (iii) TG RL also improves the quality of reasoning traces. ReaL-TG-4B achieves comparable or better levels of faithfulness, logical consistency, and alignment than G1-7B. This highlights the importance of how the environment is designed: how the reasoning problem is formulated, what temporal dependencies the model must act within, how training data is curated, and how the reward incentivizes good forecasting behavior over evolving timestamped interactions.

**Qualitative Analysis.** To further illustrate the temporal-specific challenges and why they cannot be learned through static graph training alone, we provide a case study below showing a concrete example where Qwen3-4B and G1-7B fail but ReaL-TG-4B succeeds, particularly in handling time-dependent interaction patterns and forecasting based on temporal context. We adopt the form presented in Appendix J and call this case Case 3. From Fig. 12 and 13, we can see that: (i) Qwen3-4B exhibits low-quality reasoning, repeatedly engaging in self-reflection instead of taking concrete steps toward deriving the answer; (ii) although G1-7B can list historical interactions, it relies entirely on counting the frequency of past neighbors of the query node and shows somehow lazy reasoning towards the temporal scope. In contrast, our method enables the model not only to consider node frequency but also to make decisions by explicitly checking the recency of interactions. As a result, it avoids hallucinating spurious nodes and delivers more accurate predictions, enabled by our curated data and carefully designed reward. This further proves that, during RL fine-tuning on static graph data, the model cannot effectively learn how to prioritize temporal reasoning in TG link forecasting, where temporal reasoning is critical and serves as a unique challenge.

| ReaL-TG-4B | Qwen3-4B |
|---|---|
| <think> | <think> |
| Okay, let's tackle this problem. The user is asking for the most likely destination node for a query where the source node is 1206 and the timestamp is 2,670,969. They provided a list of historical interactions between nodes with their timestamps. | Okay, let's tackle this problem. The user wants to predict the most likely destination nodes for the source node 1206 at timestamp 2670969 based on the historical interactions provided. |
| **First, I need to look at all the historical interactions involving node 1206 and see if there's a pattern or frequent destinations.** The latest interaction with node 1206 is at timestamp 2,670,884 (for destination 8852)... So, the most recent interaction is between 1206 and 8852. | First, I need to look at all the historical interactions involving source node 1206. Let me go through the list and note down each entry where the source is 1206. |
| **But maybe there are other interactions between 1206 and other nodes that are more frequent? Let me check all the entries where the source node is 1206.** | Looking at the data, here are the entries with source 1206: |
| Looking at the list: | - (1206, 8734, 2481811) |
| - (1206, 8852, 2670794) | - (1206, 8852, 2481993) |
| - (1206, 8852, 2670884) | - (1206, 8734, 2484302) |
| **So, the only interaction involving 1206 as the source is with node 8852, at timestamps 2,670,794 and 2,670,884... So, the answer is just node 8852.** | **- (221, 9149, 2489612)** |
| </think> | - (1206, 8734, 2498082) |
| <answer> | **Wait, no, that's not in the list. Let me check again...** |
| [8852] | **Wait, no. Let me go through each entry step by step...** |
| </answer> | **Wait, no. Wait, let me check again. The user provided a long list. Let me parse it correctly. Looking at the list again, the entries with source 1206:...** |
| | **Wait, no, that's not correct. Let me go through the list again. Looking at the list:...** |
| | **Wait, no, that's not correct. Let me check again. The original list is:...** |
| | **no, I think I need to check again. Let me look at the original data again...** |
| | **no, that's not in the list. Wait, maybe I'm miscounting. Actually, the original data includes:...** |
| | **no, I think I'm missing some entries. Let me check the original data again...** |
| | Wait, I think I'm getting confused. Let me go through the entire list again... |
| | Now, based on the historical interactions, what are the most common destination nodes for source 1206? Looking at the historical data for source 1206: |
| | - 8734 appears multiple times. |
| | - 8852 appears multiple times. |
| | - 9149 appears multiple times. |
| | I think I need to stop here and proceed with the data I have. Assuming that the destination nodes for source 1206 are 8734, 8852, and 9149, and **given that the query is at timestamp 2670969, which is after the last interaction with 8852 at 2670884, but before the next interactions with 8852 at 2670794, which is earlier...** |
| | But this is just a guess. Alternatively, if the user has more entries, but I'm not seeing them. However, the query is at 2670969, which is after the last interaction with 8852 at 2670884, so the next possible destination could be based on the previous interactions. |
| | Given that, the most likely destinations are 8734, 8852, and 9149. |
| | </think> |
| | <answer> |
| | [8734, 8852, 9149] |
| | </answer> |

Figure 8: Comparison of reasoning traces on the same evaluation data generated by ReaL-TG-4B and Qwen3-4B on Case 2. The **black bold** part indicates the effective reasoning strategy used by ReaL-TG-4B to obtain the correct prediction, whereas the **red bold** part marks the low-quality reasoning produced by Qwen3-4B. Qwen3-4B is prone to hallucinate incorrect links in the context graph and becomes entangled in iterative self-reflection, consuming many tokens without making substantive progress. Ultimately, it abandons the reasoning process and resorts to guessing answers.

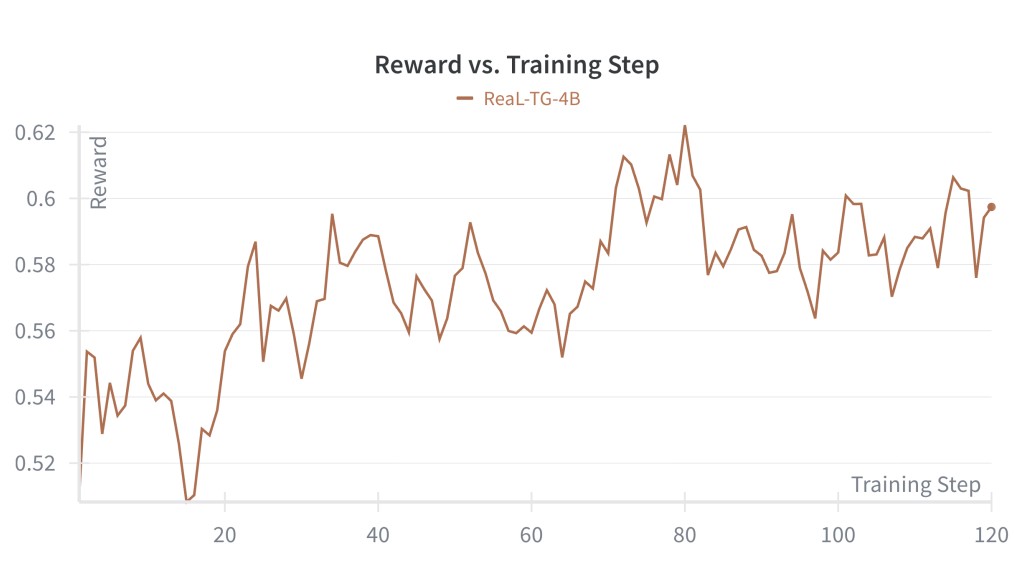

Figure 9: Reward vs. Training Step.

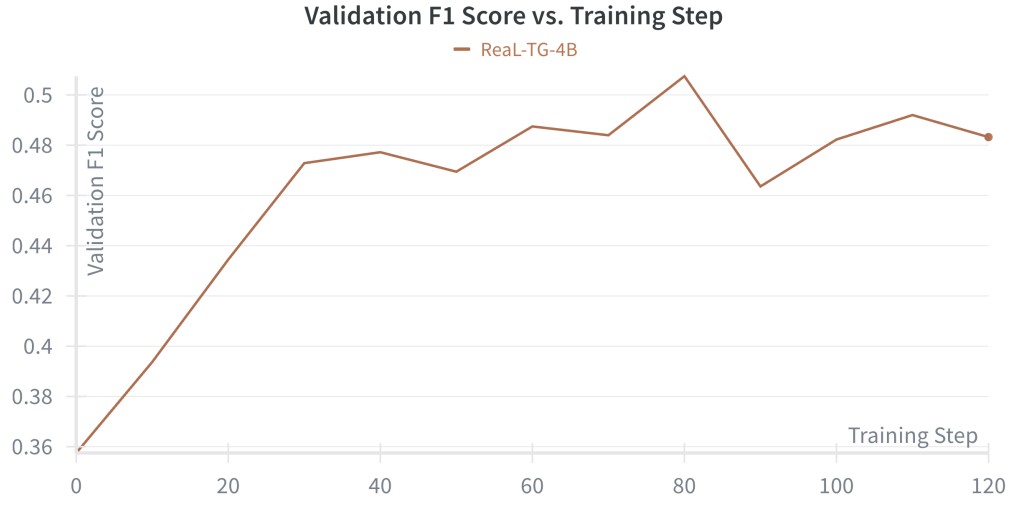

Figure 10: Validation F1 Score vs. Training Step.

```
<|system|>
You are an impartial evaluator. Your task is to inspect the model's output trajectory and determine whether it exhibits
the following forbidden phenomenon:

**Forbidden Phenomenon:**
The model justifies its prediction by claiming or implying that a fact of the form (u, v, t) or any event/interaction has
already been seen, is known to have occurred, or is available at or after the prediction timestamp.

**Allowed Behavior:**
The model is allowed to mention the data at the prediction timestamp when it refers to the query, as long as the query
data is not directly used to reach the prediction.

**Input Format:**
You will receive a JSON containing:
- `prediction timestamp`: The timestamp at which the model is making its prediction
- `output trajectory`: The model's reasoning and final answer

**Output Format:**
Return ONLY a JSON object with:
{
"Existence": 1 or 0,
"Rationale": "Brief explanation of your judgment"
}

- `Existence` = 1 if the forbidden phenomenon is present
- `Existence` = 0 if the output is valid (no temporal leakage)
- `Rationale` should cite specific phrases or reasoning patterns that led to your judgment

<|user|>
### JSON Schema
Your output must be a single JSON object that validates against this schema:
{schema_json}

### Input Data
**Prediction Timestamp:** {prediction_timestamp}

**Model Output Trajectory:**
{output_trajectory}

### Task
Analyze the output trajectory and determine if it contains the forbidden phenomenon (claiming access to information at or
after the prediction timestamp).
Return ONLY valid JSON matching the schema above.
```

Figure 11: Prompt for reward hacking quantification.

| Case3 | Qwen3-4B |
|---|---|
| $\mathcal{G}_c$:
(393, 3011, 3082176), (1582, 3011, 4275922), (1285, 3090, 4770778), (1285, 3090, 4770845), (1285, 3090, 4771892), (393, 3090, 4884301), (393, 2246, 4982280), (393, 3090, 5039720), (393, 3011, 5055435), (220, 3586, 5059255), (393, 3090, 5059943), (393, 3090, 5127655), (393, 3586, 5216901), (429, 3090, 6314559), (393, 3090, 6329062), (1111, 3090, 6343072), (1713, 3246, 6417904), (1713, 3246, 6418008), (1713, 3246, 6418451), (431, 3246, 6427011), (1713, 3246, 6440867), (431, 3246, 6450718), (1723, 3090, 6451357), (393, 3246, 6453971), (644, 3090, 6500223), (394, 3090, 6515547), (1406, 3090, 6536263), (1111, 3090, 6691162), (394, 3090, 6694175), (1647, 2093, 6770457), (63, 2093, 6796367), (393, 3090, 6803143), (1713, 2149, 6855831), (32, 2149, 6858058), (32, 2149, 6858763), (32, 2149, 6859142), (32, 2149, 6859297), (1730, 2149, 6861197), (1111, 2149, 6865512), (1111, 2149, 6865812), (393, 2149, 6873536), (233, 2093, 6964152), (63, 2093, 6967480), (1434, 2093, 7370377), (843, 2093, 7408145), (233, 2093, 7457407), (843, 2093, 7480675), (233, 2093, 8002100), (233, 2093, 8007001), (118, 2093, 8261303), (1362, 2093, 9277536), (233, 2093, 9452320), (393, 2093, 9565758), (1783, 3485, 9639975), (105, 3485, 9641586), (32, 3185, 9720300), (468, 3485, 9881207), (1127, 3485, 10046490), (1771, 3185, 10149715), (1509, 2093, 10236571), (1792, 3185, 10242323), (1771, 3185, 10249568), (1792, 3185, 10259113), (1771, 3185, 10312937), (1792, 3185, 10324933), (101, 2093, 10439919), (318, 3185, 10469910), (1786, 3485, 10598943), (1792, 3185, 10659215), (1190, 3185, 10755761), (1792, 3185, 10756968), (342, 3485, 10759547), (27, 3185, 10923721), (1332, 3485, 10944026), (1565, 2093, 11087985), (393, 3185, 11173502), (105, 3485, 11197989), (105, 3485, 11198130), (393, 3185, 11204782), (1332, 3485, 11207346), (1280, 3485, 11209898), (27, 3185, 11234534), (393, 3185, 11287360), (1582, 3485, 11371211), (1669, 3485, 11523875), (1669, 3485, 11555918), (393, 2093, 11757190), (393, 3485, 12890047), (393, 2093, 14389391), (393, 2093, 15001928)
*q*: (393, ?, 16727038)
$\{v_q\}$: {2093} | <think>
Okay, let's tackle this problem... First, I need to look at all the historical interactions involving source node 393... Looking at the list, the source node 393 has several interactions. Let me go through them one by one...
**Wait, but the query timestamp is 16727038. Let me check the latest interactions before that...**
**Looking at the list, the latest interactions for 393 before 16727038 are... Wait, but the query timestamp is 16727038, which is way later than the last historical timestamp...**
Looking at the list, the last entry in the historical data is... But the query timestamp is 16727038, which is after that.
**Wait, but in the given data, the latest timestamp is 15001928. However, the query timestamp is 16727038... Therefore, the user is asking for predictions based on all historical interactions before 16727038.**
**But in the given data, the last interaction is 15001928. Therefore, the answer must be based on that.**
**But wait, maybe I made a mistake in reading the data. Let me check again...**
So, the task is to look at all the historical interactions where source is 393, and see which destination nodes are most frequent or have some pattern.
Looking at the list, the destination nodes for 393 are: 3011, 3090, 2246, 3586, 3246, 2149, 2093, 3185, 3485.
**But looking at the counts... Now, counting the number of times 393→3090 occurs. Let's go through the list:... Now, count the occurrences of 393→3090:...**
**Looking at the list.. Let's count them:... Let me check: Looking at the list... So, the most frequent destination is 3090, then 2093, then 3185, then 3586, 3246, etc.**
**But the query timestamp is 16727038, which is after the last historical interaction... So, the model has to predict based on the historical data.**
But since 3090 is the most frequent destination for 393, it's likely that 3090 is the next destination. However, there might be other nodes... **Therefore, the prediction should be based on the frequency of previous destinations.**
So, the most likely destinations are... **So, since all the destinations that 393 has interacted with are possible, the answer should include all of them... So yes, those are all. So the answer should be a list of these sorted. Therefore, the final answer is [2093, 2149, 2246, 3011, 3090, 3185, 3246, 3485, 3586].**
</think>
<answer>
[2093, 2149, 2246, 3011, 3090, 3185, 3246, 3485, 3586]
</answer> |

Figure 12: Left: The verbalized temporal context graph, query, and ground truths in Case 3. Right: The output of Qwen3-4B. The **red bold** part marks the low-quality reasoning. Qwen3-4B is prone to be entangled in iterative self-reflection, consuming many tokens without making substantive progress. It finally reaches the answer without considering temporal dependencies.

**G1-7B**

To determine the most likely 'Query Destination Node' for 'Query Source Node' 393 at 'Query Timestamp' 16727038, we first need to analyze the historical interactions involving node 393 and identify patterns or common destinations.

Given the list of interactions, we can observe the following interactions involving node 393: - (393, 3011, 3082176) - (393, 3090, 4884301) - (393, 2246, 4982280) - (393, 3090, 5039720) - (393, 3090, 5055435) - (393, 3586, 5216901) - (393, 3090, 6329062) - (393, 3090, 6803143) - (393, 2149, 6873536) - (393, 2093, 9565758) - (393, 3185, 11173502) - (393, 3485, 12890047) - (393, 2093, 14389391) - (393, 2093, 15001928)

**Since the 'Query Timestamp' is 16727038, we need to find the most common destination nodes of node 393 that are also present in the interactions before this timestamp. However, none of the interactions listed above have a timestamp close to 16727038... Analyzing the interactions: - Node 3090 appears multiple times with node 393. - Node 3090 is the most common destination node for node 393. – Other nodes like 3011, 3586, 2149, 2093, 3185, and 3485 also appear, but not as frequently as 3090.**

Given the data, **the most likely destination node for node 393 at any given time is 3090, as it appears the most frequently in the interactions.**

Therefore, the most plausible 'Query Destination Node' for 'Query Source Node' 393 at 'Query Timestamp' 16727038 is:

<answer>
[3090]
</answer>

**ReaL-TG-4B**

<think>
Okay, let's tackle this problem. The user is asking for the most likely destination node for a query where the source node is 393 and the timestamp is 16,727,038. They provided a list of historical interactions between nodes with their timestamps.

First, I need to look at all the historical interactions involving node 393 and see if there's a pattern or frequent destinations. The latest interaction with node 393 is at timestamp 15,001,928 (for the entry (393, 2093, 15001928)). The query timestamp is 16,727,038, which is after that, so we only consider up to that point.

Looking at the historical data, I'll filter the entries where the source node is 393. The relevant entries are: 1. (393, 2093, 14389391) 2. (393, 2093, 15001928)

Wait, the second entry is (393, 2093, 15001928). So, **the most recent interaction between 393 and 2093 at timestamp 15,001,928.** The query timestamp is 16,727,038, so 15 million is before that. So, **the latest interaction is with node 2093.** But maybe there are other interactions where source is 393? Let me check again. The list provided might have other entries where source is 393. Let me scan through the list again.

Looking at the list again, there's another entry: (393, 3485, 12890047), and (393, 2093, 14389391), and (393, 2093, 15001928). So, another one at (393, 3485, 12890047). So, two interactions: one with 2093 at two different timestamps (14,389,391 and 15,001,928), and another with 3485 at 12,890,047. So, the source 393 has interacted with 2093 twice and with 3485 once. So, **the most frequent destination is 2093.** But maybe there are others? Let me check again. The other entries where source is 393: maybe the entry (393, 3485, 12890047). **So, total of three interactions: two with 2093 (at 14.3m and 15m) and one with 3485 (at 12.8m). So, the answer would be 2093.**
</think>
<answer>
[2093]
</answer>

Figure 13: Comparison of reasoning traces generated by G1-7B and ReaL-TG-4B on Case 3. The **black bold** part indicates the effective reasoning strategy used by ReaL-TG-4B to obtain the correct prediction, whereas the **red bold** part marks the misleading reasoning produced by G1-7B. G1-7B fails to show how to prioritize temporal reasoning and shows somehow lazy reasoning towards the temporal scope, while ReaL-TG-4B considers both frequency and temporal recency as a whole in decision making.

