# OpenReview forum: "Self-Exploring Language Models for Explainable Link Forecasting on Temporal Graphs via Reinforcement Learning"
_ICLR.cc/2026/Conference — ICLR 2026 Conference Withdrawn Submission_

### Official Review · Reviewer_xRzV · 2025-10-20

**Soundness:** 3
**Presentation:** 3
**Contribution:** 3
**Rating:** 8
**Confidence:** 4

**Summary:**

The paper proposes an approach for future link prediction in temporal graphs (called link forecasting). It finetunes an LLM (Qwen3-4B) for this task via reinforcement learning. All textual data is removed from the graphs and replaced by IDs.

**Strengths:**

- Original idea of using LLMs for link forecasting
- Strong results compared both to other LLMs and compared to TGNNs
- Explanation traces are provided and evaluated in a user study
- Paper is well-structured

**Weaknesses:**

- The motivation for using an LLM remains unclear (apart from some success in preliminary work)

**Questions:**

- What is the motivation of using an LLM if all textual data is removed?
- Do you have a hypothesis why your approach works so well despite removing all textual data?

---

> ### Author Response · Authors · 2025-11-21
> **Response to Reviewer xRzV**
>
> We thank the reviewer for the time and effort. Here is our response.
>
> > W1: The motivation for using an LLM remains unclear (apart from some success in preliminary work)
>
> We respectfully clarify that our motivation for using LLMs is grounded in the submission. Forecasting is not only about predictive accuracy but also about understanding why a prediction is made. Traditional TGNN-based methods cannot provide human-readable reasoning and thus offer limited transparency (lines 46–47). LLMs, in contrast, can naturally produce textual explanations (lines 52–55), which allows us to treat link forecasting as a question-answering problem where predictions are supported by explicit reasoning. This explainability is important in forecasting applications, where decision-makers must be able to assess and trust the rationale behind a model’s outputs.
> We also note that LLMs offer practical advantages that complement traditional TGNNs. Recent findings show that LLMs possess emerging zero-shot reasoning and generalization abilities (lines 52–55), which motivates studying whether such capabilities can be leveraged for temporal graph forecasting. Unlike TGNNs that require retraining for each new temporal graph (lines 47–48), LLMs can adapt to new graphs through prompting alone. This creates a promising direction for building flexible and explainable forecasting models that generalize across diverse temporal graphs.
>
> > Q1: What is the motivation of using an LLM if all textual data is removed?
>
> As noted in our submission (lines 135–137), our goal is not to evaluate how LLMs interpret textual attributes, which has already been extensively studied in prior works. Instead, we intentionally remove textual content to isolate and examine the LLM’s ability to reason over temporal dependencies and graph topologies. This design allows us to focus on temporal and structural reasoning rather than relying on information contained in text attributes.
>
> This choice is important because when abundant textual information is available, predictions may be driven by text-derived cues rather than by reasoning over temporal signals or graph structure. Our work aims to understand whether LLMs can handle these two core forms of reasoning that are fundamental to temporal graph forecasting, and how we can improve them. We believe this complements existing studies that focus on reasoning from textual meaning. To complement, we refer the reviewer to our response to W1 to have a more comprehensive understanding.
>
> > Q2: Do you have a hypothesis why your approach works so well despite removing all textual data?
>
> Our hypothesis is based on two observations. First, recent studies suggest that LLMs possess non-trivial abilities to reason over temporal sequences and simple static graph structures [1,2,3] even without textual attributes, and our own preliminary experiments confirmed this trend. We observed that LLMs can identify certain recurring temporal and structural patterns and use them to reach reasonable forecasting labels. This indicates that, beyond semantic understanding, LLMs retain some inherent capacity to operate on temporal signals and graph topologies alone.
>
> Second, although these initial capabilities exist, we also observed that LLMs often produce inconsistent or unfaithful reasoning trajectories when applied to temporal graph forecasting at scale (we have some examples in our qualitative analysis in Appendix J). Motivated by recent progress showing that RL fine-tuning can strengthen reasoning in LLMs, we explored whether such techniques could help LLMs form more stable reasoning over temporal dependencies and graph structures. Our experimental results support this hypothesis, as the model becomes more consistent and accurate in both prediction and reasoning when applied in this setting.
>
> [1] Fons, E., Kaur, R., Palande, S., Zeng, Z., Balch, T., Veloso, M., & Vyetrenko, S. (2024, November). Evaluating Large Language Models on Time Series Feature Understanding: A Comprehensive Taxonomy and Benchmark. In Proceedings of the 2024 Conference on Empirical Methods in Natural Language Processing (pp. 21598-21634).
>
> [2] Fatemi, B., Halcrow, J., & Perozzi, B. Talk like a Graph: Encoding Graphs for Large Language Models. In The Twelfth International Conference on Learning Representations.
>
> [3] Huang, S., Parviz, A., Kondrup, E., Yang, Z., Ding, Z., Bronstein, M., ... & Rabusseau, G. (2025). Are Large Language Models Good Temporal Graph Learners?. arXiv preprint arXiv:2506.05393.

---

### Official Review · Reviewer_VHSj · 2025-10-23

**Soundness:** 3
**Presentation:** 3
**Contribution:** 2
**Rating:** 2
**Confidence:** 4

**Summary:**

This paper constructs temporal graphs in natural language and finetunes LLMs with reinforcement learning on the dataset. The finetuned model outperforms larger LLMs under the proposed pMRR and LLM-as-a-Judge metrics.

**Strengths:**

The paper presents that the reasoning ability of LLMs on temporal graphs can be enhanced by RL training, which is not explored by previous works of LLM on temporal graphs, which mainly focus on evaluation and ICL. The full pipeline, from data construction to training and evaluation, is reasonable and easy to follow.

**Weaknesses:**

The main weakness is a lack of innovation and further insights. The idea that the graph reasoning ability of LLMs can be incentivized through reinforcement learning has been illustrated in previous work. Although the authors claim that previous works focus on static graphs while they target temporal graphs, they don't show what difference it will bring to the reasoning or further insights (e.g., what is special for temporal graphs but not static graphs). In the whole pipeline, the innovation is quite limited. For example, the algorithm is the well-established GRPO, and the LLM-as-Judge evaluation is also commonly used.

**Questions:**

1. The base models only include Qwen3-4B and Qwen3-0.6B. What's the performance of the model finetuned from a larger Qwen3-8B?

2. What's the performance of the ReaL-TG models on OOD benchmarks, e.g., unseen temporal graph benchmarks?

3. Does the training harm or enhance the general reasoning ability of base models, e.g., the math benchmarks?

4. How does the RL training increase the reasoning of LLMs on temporal graphs? Can you show some intuitive examples?

5. The reason for proposing pMRR is to evaluate the over-generation of LLMs. Can the recall in F1 play a similar role?

6. Why don't you use the proposed metrics as rewards during RL training?

---

> ### Author Response · Authors · 2025-11-21
> **Response to Reviewer VHSj: Part A**
>
> Thank you for your time and effort for reviewing our paper. Here is our response. We hope it will maximally address your concerns.
>
> > W1: lack of innovation and further insights
>
> We understand your concern. We agree that ReaL-TG builds based on existing components, however, the novelty of our work lies not in re-introducing these techniques, but in **how they are adapted and applied to answer a new research question: whether and how RL can improve LLMs in explainable forecasting over temporal graphs based on temporal graph topologies.** Our work is the first to explore RL verifiable rewards (RLVR) for enhancing LLM performance specifically on temporal graph link forecasting.
>
> While ReaL-TG adopts the general LLM-as-a-Judge paradigm, **our evaluation method is substantially adapted to this domain.** We design task-specific evaluation rubrics and scoring criteria tailored to link forecasting, aligned with the evaluation protocol described in our paper, such as assessing whether the model reasoning induces factual hallucinations, logical inconsistency hallucinations and justification hallucinations. Existing judge protocols for math, code, or generic QA cannot assess these aspects. We also validate our judge outputs against human annotations to ensure reliability.
>
> We believe standard machine learning methods are designed to be reused, and applying them effectively to a new domain often requires substantial methodological adaptation. In our case, we carefully designed a task-specific reward, collected diverse RL training data across multiple temporal graphs, and constructed a dedicated rubric for evaluating the reasoning quality of LLM in explainable forecasting. These steps are non-trivial and necessary to make RL function meaningfully in this domain. We fully respect your perspective and hope that this clarification helps provide a clearer and more positive impression of our work.
>
> > W1: What is special for temporal graphs but not static graphs?
>
> In contrast to static graphs, temporal graphs introduce challenges that fundamentally alter the reasoning process. Static graph reasoning requires understanding a fixed topology. Temporal graph reasoning, however, requires the model to process **dynamically evolving structures**: new neighbors may emerge, older neighbors may become less relevant, and the importance of edges shifts over time. Moreover, temporal dependencies must be understood, such as “A interacts with B, then B interacts with C, which increases the likelihood of A interacting with C shortly after.” These dynamics do not exist in static graph settings. Thus, transferring RL-based reasoning from static to temporal graphs is not straightforward; it requires adapting the pipeline so the model can reason about both the **time-varying topology and the temporal ordering of node interactions**, which is central to forecasting.
>
> Furthermore, our work focuses on forecasting, where future interactions do not have timely ground-truth labels in real-world deployment. This makes explainable reasoning especially important, since users must rely on the model’s justification rather than immediate accuracy feedback. Prior LLM-based static graph work mainly optimizes for accuracy on fully labeled data, but forecasting requires models to provide transparent reasoning to ensure trustworthiness when labels are not yet available.
>
> > Q1: The base models only include Qwen3-4B and Qwen3-0.6B. What's the performance of the model finetuned from a larger Qwen3-8B?
>
> We thank the reviewer for the suggestion. In our compute environment, due to limited budget, we only have access to 4 × 80 GB H100 GPUs. Under the GRPO hyperparameters used in our experiments (16,384 maximum output length, batch size 32, mini-batch size 16, and 5 rollouts per prompt; see Appendix D), the 4B model already occupies around 75 GB of VRAM on each GPU during training. GRPO requires keeping both the policy and reference models as well as multiple rollout activations in memory. Scaling this setup from a 4B to an 8B model would exceed the available VRAM per GPU in our 4-card configuration, making full GRPO fine-tuning of an 8B model infeasible in our setting.
>
> Although we cannot include the 8B variant, the comparison between Qwen3-0.6B and Qwen3-4B already provides a clear conclusion. The 4B model consistently benefits from our method and shows substantial improvement over its base model, while the 0.6B model demonstrates the same trend but at a smaller scale. These two results together confirm our main finding that enhancing the reasoning process leads to consistent gains in temporal graph forecasting across model sizes.

---

> > ### Author Response · Authors · 2025-11-21
> > **Response to Reviewer VHSj: Part B**
> >
> > > Q2: What's the performance of the ReaL-TG models on OOD benchmarks, e.g., unseen temporal graph benchmarks?
> >
> > **We would like to note that the requested OOD results are already included in our submission, specifically in Table 2 as main results. We kindly ask the reviewer to revisit this part of the paper if you have missed them while reviewing, as it directly addresses the concern.** In Table 2, we evaluate ReaL-TG on two unseen temporal graph datasets, uci and enron, which the model has never encountered during training. These results show clear improvements over the Qwen3-4B base model on both OOD benchmarks, demonstrating that ReaL-TG generalizes effectively to unseen temporal graphs. This finding highlights an important contribution of our work, showing that the enhanced reasoning abilities acquired by ReaL-TG transfer beyond the training graph and support robust cross-graph forecasting.
> >
> > > Q3: Does the training harm or enhance the general reasoning ability of base models, e.g., the math benchmarks?
> >
> > We are currently experiencing a server breakdown, and we need to wait for it to be restored before running the additional experiments. We will notify you as soon as the results are ready. We hope you could understand and we kindly invite you to first consider our responses to other points.
> >
> > > Q4: How does the RL training increase the reasoning of LLMs on temporal graphs? Can you show some intuitive examples?
> >
> > **We have already provided qualitative case studies in Appendix J during submission. Please check if you have missed them while reviewing.** In Case 1 (Fig. 6 and 7), after RL the model no longer exhausts the context window by repeating content; instead, it identifies the most plausible destination node by correctly leveraging interaction recency. In Case 2 (Fig. 6 and 8), the RL-trained model is less likely to fall into iterative self-reflection and shows greater confidence and effectiveness in applying reasoning strategies to support its predictions. Overall, exploration during RL is essential for improving both prediction accuracy and reasoning quality. While the base model can already produce plausible reasoning, it still needs RL to learn how to adapt its reasoning skills to the specific context.
> >
> > > Q5: The reason for proposing pMRR is to evaluate the over-generation of LLMs. Can the recall in F1 play a similar role?
> >
> > Unfortunately no. Recall is defined as true positives divided by true positives plus false negatives, so a model that predicts all candidates as positive will always achieve recall equal to 1 regardless of how many false positives it produces. Over-generation is instead reflected in the number of false positives, which recall does not take into account.
> >
> > > Q6: Why don't you use the proposed metrics as rewards during RL training?
> >
> > We do not use our proposed pMRR as reward during RL training because metrics such as MRR and pMRR depend heavily on dataset-specific statistics, including the number of candidate nodes or negative edges. This dependence makes the reward scale unstable across different temporal graphs.
> >
> > To illustrate this, consider a setting with exactly one positive node among $N$ total nodes. Under a uniform random ranking, the rank of the positive node is uniformly distributed over $\{1,\dots,N\}$. The expected MRR is therefore
> > $\mathbb{E}[\text{MRR}] = \mathbb{E}\left[\frac{1}{\text{rank}}\right] = \frac{1}{N}\sum_{k=1}^{N}\frac{1}{k} = \frac{H_N}{N}$,
> > where $H_N$ is the $N$-th harmonic number. For $N=10$ nodes, this gives $\mathbb{E}[\text{MRR}] = H_{10}/10 \approx 0.29$. For $N=1000$ nodes, this drops to $H_{1000}/1000 \approx 0.0075$.
> > These examples show that MRR’s scale shrinks rapidly as the number of candidate nodes increases, even though the model behavior (uniform random guessing) is unchanged. The same characteristic applies to pMRR.
> >
> > Since MRR/pMRR varies with the size of the underlying temporal graph rather than with model quality alone, using it as a reward would introduce dataset-dependent biases. Our goal is to train a model that learns forecasting behavior from the temporal context itself, without encoding or exploiting global graph statistics. Therefore, we do not use MRR-like metrics as rewards during RL training.

---

> > > ### Author Response · Authors · 2025-11-24
> > > **Response to Reviewer VHSj: Part C**
> > >
> > > Hi, we hope you had a nice weekend.
> > >
> > > As requested in
> > > > Q3: Does the training harm or enhance the general reasoning ability of base models, e.g., the math benchmarks?
> > >
> > > we have just obtained additional experimental results for Qwen3-4B and ReaL-TG-4B on GSM8K and MATH-500, two standard benchmarks for evaluating the mathematical reasoning capabilities of LLMs. The metric used for both datasets is accuracy. Both models are evaluated with identical hyperparameters, including temperature set to 0.6, top-p set to 0.95, and top-k set to 20, as recommended by the official Qwen3 technical report. We set the maximum output length to 30,000 tokens to ensure that long-form reasoning is not truncated before producing a final answer. For evaluation, we used EvalScope, a widely used LLM evaluation framework on GitHub with more than 2k stars, which provides standardized and reproducible evaluation implementations. MATH-500 includes 5 level of problems, so here we also include level-wise results. The results are presented in the following table.
> > >
> > > | Dataset   | Subset   | Num  | Qwen3-4B | ReaL-TG-4B |
> > > |-----------|----------|------|----------|------------|
> > > | gsm8k     | main     | 1319 | 0.9484   | 0.9492     |
> > > | math_500  | Level 1  | 43   | 0.9535   | 0.9767     |
> > > | math_500  | Level 2  | 90   | 0.9778   | 0.9778     |
> > > | math_500  | Level 3  | 105  | 0.9714   | 0.9524     |
> > > | math_500  | Level 4  | 128  | 0.9531   | 0.9609     |
> > > | math_500  | Level 5  | 134  | 0.9030   | 0.9104     |
> > > | math_500  | OVERALL  | 500  | 0.9480   | 0.9500     |
> > >
> > > We observe that our fine-tuned model does not sacrifice mathematical reasoning capabilities. In fact, it even shows slight improvements in most settings. We believe the reason is the following. During training, we explicitly impose a KL divergence-based loss between the reference model, which is the original Qwen3, and the fine-tuned model. By inspecting our training logs, we found that at the final training step the KL loss was as small as 0.00025. This indicates that our model has not drifted far from the original Qwen3, and therefore should not behave very differently in domains beyond our training target, namely, temporal graph link forecasting.
> > >
> > > At the same time, we do not want to overclaim. We are not arguing that our training procedure improves general mathematical reasoning because the observed gains are small and not the focus of our method. Our motivation is solely to enhance LLMs for link forecasting, and broader gains are outside our scope. The purpose of this comparison is simply to provide empirical evidence that our fine-tuning does not harm the model’s general reasoning ability. We hope this addresses your concern.

---

> ### Comment · Reviewer_VHSj · 2025-11-24
>
> Thank you for your detailed response, which addresses most of my concerns. I will raise my rating accordingly.
>
> However, my primary concern remains. The paradigm of "collecting data → encoding into natural language → training with GRPO → evaluation" has proven effective on static graphs in prior works. While you rightly note that temporal graphs have dynamically evolving structures and require an understanding of temporal dependencies, I am not yet convinced that this introduces fundamentally new challenges to the core methodology, particularly the training process.

---

> > ### Author Response · Authors · 2025-11-28
> > **Revision 2.0 Online; Looking Forward to Feedback**
> >
> > Hi,
> >
> > We have updated our PDF to version 2.0, which now includes additional experiments on math datasets, a detailed comparison with G1, an extended case study, and a discussion clarifying why temporal graph reasoning is fundamentally different from static graph reasoning—and why models trained solely on static graph data cannot effectively perform temporal reasoning. These additions directly correspond to the points raised in your review, and we would greatly appreciate it if you could take a moment to look through them. We hope our efforts adequately address your concerns.
> >
> > We are fully aware that concurrent works such as G1 also explore graph problems using RL. Our comparison is not intended to suggest that static graph-focused methods are “weaker” or not “well-rounded”, but rather to highlight that the core challenges in temporal graph reasoning are distinct, particularly when dealing with large, real-world TGs, as in our setting.
> >
> > We wish you a pleasant weekend in advance and look forward to your feedback.

---

> ### Author Response · Authors · 2025-11-26
> **Additional Experiments Showing Challenge of our Task: Part A**
>
> We totally understand your concern. Here we provide additional experiments to help to better clarify the challenge of temporal graph forecasting beyond static graph reasoning.
>
> G1 [1] is a notable concurrent work that demonstrates the effectiveness of RL fine-tuning on static graph reasoning tasks at scale (100k training examples). To evaluate whether such large-scale static graph training is sufficient for temporal graph forecasting, we tested G1-3B and G1-7B (checkpoints provided in their original work) on our evaluation set using their default hyperparameters, and compare them Qwen3-4B and our ReaL-TG-4B.
>
> | Dataset | wiki MRR | wiki pMRR | subreddit MRR | subreddit pMRR | coin MRR | coin pMRR | flight MRR | flight pMRR | uci MRR | uci pMRR | enron MRR | enron pMRR | Overall MRR | Overall pMRR |
> |---------|----------|------------|----------------|-----------------|-----------|------------|-------------|--------------|-----------|-----------|-------------|--------------|---------------|----------------|
> | Qwen3-4B   | 0.721 | 0.682 | 0.678 | 0.639 | 0.368 | 0.333 | 0.090 | 0.087 | 0.300 | 0.239 | 0.174 | 0.137 | 0.375 | 0.339 |
> | G1-3B      | 0.650 | 0.641 | 0.642 | 0.629 | 0.299 | 0.286 | 0.178 | 0.161 | 0.376 | 0.348 | 0.270 | 0.246 | 0.382 | 0.382 |
> | G1-7B      | 0.794 | 0.782 | **0.786** | **0.770** | 0.383 | 0.331 | 0.193 | 0.151 | 0.485 | 0.445 | 0.464 | 0.398 | 0.523 | 0.484 |
> | ReaL-TG-4B | **0.824** | **0.792** | 0.765 | 0.726 | **0.431** | **0.401** | **0.198** | **0.175** | **0.607** | **0.523** | **0.492** | **0.435** | **0.552** | **0.508** |
> ### Table: Prediction Accuracy
>
> | Model      | $\bar{\delta}_f$ | $\bar{\delta}_{lc}$ | $\bar{\delta}_a$ |
> |------------|------------------|----------------------|-------------------|
> | Qwen3-4B   | 0.683            | 0.700                | 0.653             |
> | G1-3B      | 0.685            | 0.692                | 0.600             |
> | G1-7B      | 0.859            | 0.872                | **0.750**         |
> | ReaL-TG-4B | **0.885**        | **0.880**            | 0.732             |
> ### Table: Reasoning Quality
>
> From these experiments, we observe a consistent pattern:
> - **Static graph RL helps, but does not solve temporal forecasting.**
> G1-3B improves over Qwen3-4B, showing that RL on graph reasoning data is indeed useful.
> However, both G1 models still lag behind ReaL-TG-4B on almost all datasets, especially on uci and enron.
> - **Even large-scale static graph RL does not transfer to temporal graphs.**
> Despite using ~100× more training instances, G1-7B still underperforms ReaL-TG-4B, which was trained on only 1k temporal graph training instances.
> This indicates that temporal graphs introduce reasoning challenges that simply do not arise in static graph settings, and thus cannot be learned from static graph data alone.
> - **Temporal graph RL also improves the quality of reasoning traces.**
> ReaL-TG-4B achieves comparable or better levels of faithfulness, logical consistency, and alignment than G1-7B.
> This highlights the importance of how the environment is designed: how the reasoning problem is formulated, what temporal dependencies the model must act within, how training data is curated, and how the reward incentivizes good forecasting behavior over evolving time-stamped interactions.
>
> **More broadly, in the reasoning-enhancement literature, algorithmic innovation is undoubtedly important. But we also believe it is equally important to “scale the environment”:
> how to define and formulate a reasoning problem properly in a domain that presents unique challenges that cannot be solved by learning from other domains; how to collect data that exposes these challenges; how to design rewards that reflect what good temporal reasoning should look like; and how to evaluate models in a way that meaningfully captures these capabilities.**
> G1 achieves this for static graphs, while our work focuses on temporal graphs, where dynamic evolution, recency, temporal causality, and changing node states create fundamentally different reasoning requirements.
>
> To further illustrate the temporal-specific challenges and why they cannot be learned through static-graph training alone, we provide a case study below showing a concrete example where G1-7B fail but ReaL-TG succeeds, particularly in handling time-dependent interaction patterns and forecasting based on temporal context. We adopt the form presented in Appendix J and call this case Case 3. Same as Case 1 and 2 in the paper, we will provide here the temporal context graph $\mathcal{G}_c$, the query $q$, the ground truth answer set {$v_q$}, and the responses of G1-7B and ReaL-TG-4B.

---

> > ### Author Response · Authors · 2025-11-26
> > **Additional Experiments Showing Challenge of our Task: Part B**
> >
> > ```
> > $\mathcal{G}_c$:
> > (393,3011,3082176) (1582,3011,4275922) (1285,3090,4770778) (1285,3090,4770845) (1285,3090,4771892)
> > (393,3090,4884301) (393,2246,4982280) (393,3090,5039720) (393,3011,5055435) (220,3586,5059255)
> > (393,3090,5059943) (393,3090,5127655) (393,3586,5216901) (429,3090,6314559) (393,3090,6329062)
> > (1111,3090,6343072) (1713,3246,6417904) (1713,3246,6418008) (1713,3246,6418451) (431,3246,6427011)
> > (1713,3246,6440867) (431,3246,6450718) (1723,3090,6451357) (393,3246,6453971) (644,3090,6500223)
> > (394,3090,6515547) (1406,3090,6536263) (1111,3090,6691162) (394,3090,6694175) (1647,2093,6770457)
> > (63,2093,6796367) (393,3090,6803143) (1713,2149,6855831) (32,2149,6858058) (32,2149,6858763)
> > (32,2149,6859142) (32,2149,6859297) (1730,2149,6861197) (1111,2149,6865512) (1111,2149,6865812)
> > (393,2149,6873536) (233,2093,6964152) (63,2093,6967480) (1434,2093,7370377) (843,2093,7408145)
> > (233,2093,7457407) (843,2093,7480675) (233,2093,8002100) (233,2093,8007001) (118,2093,8261303)
> > (1362,2093,9277536) (233,2093,9452320) (393,2093,9565758) (1783,3485,9639975) (105,3485,9641586)
> > (32,3185,9720300) (468,3485,9881207) (1127,3485,10046490) (1771,3185,10149715) (1509,2093,10236571)
> > (1792,3185,10242323) (1771,3185,10249568) (1792,3185,10259113) (1771,3185,10312937) (1792,3185,10324933)
> > (101,2093,10439919) (318,3185,10469910) (1786,3485,10598943) (1792,3185,10659215) (1190,3185,10755761)
> > (1792,3185,10756968) (342,3485,10759547) (27,3185,10923721) (1332,3485,10944026) (1565,2093,11087985)
> > (393,3185,11173502) (105,3485,11197989) (105,3485,11198130) (393,3185,11204782) (1332,3485,11207346)
> > (1280,3485,11209898) (27,3185,11234534) (393,3185,11287360) (1582,3485,11371211) (1669,3485,11523875)
> > (1669,3485,11555918) (393,2093,11757190) (393,3485,12890047) (393,2093,14389391) (393,2093,15001928)
> >
> > q = (393,?,16727038)
> > {v_q} = {2093}
> > ```
> > G1-7B:
> > ```
> > To determine the most likely `Query Destination Node` for `Query Source Node` 393 at `Query Timestamp` 16727038, we first need to analyze the historical interactions involving node 393 and identify patterns or common destinations.
> >
> > Given the list of interactions, we can observe the following interactions involving node 393:
> > - (393, 3011, 3082176)
> > - (393, 3090, 4884301)
> > - (393, 2246, 4982280)
> > - (393, 3090, 5039720)
> > - (393, 3090, 5055435)
> > - (393, 3586, 5216901)
> > - (393, 3090, 6329062)
> > - (393, 3090, 6803143)
> > - (393, 2149, 6873536)
> > - (393, 2093, 9565758)
> > - (393, 3185, 11173502)
> > - (393, 3485, 12890047)
> > - (393, 2093, 14389391)
> > - (393, 2093, 15001928)
> >
> > *Since the `Query Timestamp` is 16727038, we need to find the most common destination nodes of node 393 that are also present in the interactions before this timestamp. However, none of the interactions listed above have a timestamp close to 16727038…*
> >
> > *Analyzing the interactions:
> > - Node 3090 appears multiple times with node 393.
> > - Node 3090 is the most common destination node for node 393.
> > – Other nodes like 3011, 3586, 2149, 2093, 3185, and 3485 also appear, but not as frequently as 3090. *
> >
> > Given the data, *the most likely destination node for node 393 at any given time is 3090,*
> > *as it appears the most frequently in the interactions.*
> >
> > Therefore, the most plausible `Query Destination Node` for `Query Source Node` 393 at `Query Timestamp` 16727038 is:
> > <answer>[3090]</answer>
> > ```

---

> ### Author Response · Authors · 2025-11-26
> **Additional Experiments Showing Challenge of our Task: Part C**
>
> ReaL-TG-4B:
> ```
> <think>
> Okay, let's tackle this problem. The user is asking for the most likely destination node for a query where the source node is 393 and the timestamp is 16,727,038. They provided a list of historical interactions between nodes with their timestamps.
>
> First, I need to look at all the historical interactions involving node 393 and see if there's a pattern or frequent destinations. The latest interaction with node 393 is at timestamp 15,001,928 (for the entry (393, 2093, 15001928)). The query timestamp is 16,727,038, which is after that, so we only consider up to that point.
>
> Looking at the historical data, I'll filter the entries where the source node is 393. The relevant entries are:
>
> 1. (393, 2093, 14389391)
> 2. (393, 2093, 15001928)
>
> Wait, the second entry is (393, 2093, 15001928). So,** the most recent interaction between 393 and 2093 at timestamp 15,001,928.** The query timestamp is 16,727,038, so 15 million is before that. So, **the latest interaction is with node 2093.** But maybe there are other interactions where source is 393? Let me check again. The list provided might have other entries where source is 393. Let me scan through the list again.
>
> Looking at the list again, there's another entry: (393, 3485, 12890047), and (393, 2093, 14389391), and (393, 2093, 15001928). So, another one at (393, 3485, 12890047). So, two interactions: one with 2093 at two different timestamps (14,389,391 and 15,001,928), and another with 3485 at 12,890,047. So, the source 393 has interacted with 2093 twice and with 3485 once. So, **the most frequent destination is 2093.** But maybe there are others? Let me check again. The other entries where source is 393: maybe the entry (393, 3485, 12890047). So, total of three interactions: two with 2093 (at 14.3m and 15m) and one with 3485 (at 12.8m). So, the answer would be 2093.
> </think>
> ```
>
> Note that we have marked the related parts where G1-7B makes mistakes with \*...\*, where critical temporal reasoning is carried out by ReaL-TG-4B with \*\*...\*\*. **From this case, we observe that during RL tuning on a static graph, the model fails to learn how to prioritize temporal reasoning and show somehow lazy reasoning towards the temporal scope. Although it can list historical interactions, it relies entirely on counting the frequency of past neighbors of the query node. In contrast, our method enables the model not only to consider node frequency but also to make decisions by explicitly checking the recency of interactions. As a result, it avoids hallucinating spurious nodes and produces more accurate predictions, thanks to our training process, including curated data and carefully designed reward.**
>
> We will update our PDF again to include these analysis with better visualizations. We hope this gives you a better clarification. And please feel free to initiate further discussion. We appreciate your effort and are really grateful to your feedback.
>
> [1] Guo, X., Li, A., Wang, Y., Jegelka, S., & Wang, Y. G1: Teaching LLMs to Reason on Graphs with Reinforcement Learning. In The Exploration in AI Today Workshop at ICML 2025.

---

### Official Review · Reviewer_W9DC · 2025-10-31

**Soundness:** 2
**Presentation:** 2
**Contribution:** 2
**Rating:** 4
**Confidence:** 5

**Summary:**

This paper introduces Real-TG, which employs reinforcement learning with verified rewards (LRVR) like GRPO to fine-tune an LLM for more effective explainable link forecasting on anonymized temporal graphs. It formulates the link forecasting as the QA tasks and designs a specific prompt to instruct LLMs to identify the destination node. The authors also propose a metric (i.e., pMRR) to evaluate the prediction capability of LLMs (as LLMs can provide discouraging over-generation). Extensive experiments show that Real-TG-4B achieves good results across different vanilla LLMs and traditional link forecasting methods.

**Strengths:**

* The idea of introducing LRVR to LLM-based temporal graph models is interesting.
* The proposed fine-tuning paradigm, Real-TG, is simple and effective.
* The paper is well-written and easy to follow.

**Weaknesses:**

* The novelty of the proposed Real-TG is limited and lacks technical innovation. In Real-TG, the QA formulation [1], the GRPO pipeline [2], the $\alpha$-temporal random walk [3], and the LLM-as-a-Judge evaluation [4] are introduced from existing works. All these mechanisms resemble existing works and do not introduce principled advances in LLM-based temporal graph learning or RLVR.
* The motivation of the model component design is unclear.
	* Why is RL chosen instead of SFT? In the literature on large reasoning models, RL is typically used to optimize preferences or human-aligned objectives; however, for tasks with a single correct answer, SFT may be more appropriate. The authors should clarify why RL is necessary and what concrete benefits it confers in this setting.
	* Why adopt $\alpha$-temporal random walks rather than simpler k-hop neighborhoods?
	* Why use F1 as the reward metric rather than other ranking/IR metrics (e.g., MRR)? In equation 1, do the authors compute this F1 over a single pair of samples, since F1 fundamentally requires aggregation over multiple samples? An illustrative example may help with a better understanding of the paper.
* The GRPO procedure is known to be sensitive to hyperparameters. Does Real-TG also suffer from such instability?
* Plots showing reward vs. training step would greatly strengthen confidence in the RL training procedure.
* Why not conduct experiments in the recently proposed benchmark [5] and compare the models that also utilize LLMs for temporal graph learning [6]?

[1] Are Large Language Models Good Temporal Graph Learners?

[2] DeepSeekMath: Pushing the Limits of Mathematical Reasoning in Open Language Models

[3] Zebra: When temporal graph neural networks meet temporal personalized pagerank

[4] Judging Llm-as-a-judge with Mt-bench and Chatbot Arena

[5] DTGB: A Comprehensive Benchmark for Dynamic Text-Attributed Graphs

[6] Unifying Text Semantics and Graph Structures for Temporal Text-attributed Graphs with Large Language Models

**Questions:**

No

---

> ### Author Response · Authors · 2025-11-21
> **Response to Reviewer W9DC: Part A**
>
> We appreciate your time and effort in reviewing our paper. Here is our response.
>
> > W1: The novelty of the proposed Real-TG is limited and lacks technical innovation. In Real-TG, the QA formulation, the GRPO pipeline, the $\alpha$-temporal random walk, and the LLM-as-a-Judge evaluation are introduced from existing works. All these mechanisms resemble existing works and do not introduce principled advances in LLM-based temporal graph learning or RLVR.
>
> We understand your concern. We agree that ReaL-TG builds based on existing components, however, the novelty of our work lies not in re-introducing these techniques, but in **how they are adapted and applied to answer a new research question: whether and how RL can improve LLMs in explainable forecasting over temporal graphs based on temporal graph topologies.** Our work is the first to explore RL verifiable rewards (RLVR) for enhancing LLM performance specifically on temporal graph link forecasting.
>
> While ReaL-TG adopts the general LLM-as-a-Judge paradigm, **our evaluation method is substantially adapted to this domain**. We design task-specific evaluation rubrics and scoring criteria tailored to link forecasting, aligned with the evaluation protocol described in our paper, such as assessing whether the model reasoning induces factual hallucinations, logical inconsistency hallucinations and justification hallucinations. Existing judge protocols for math, code, or generic QA cannot assess these aspects. We also validate our judge outputs against human annotations to ensure reliability.
>
> Finally, we respectfully note that it is not very fair to classify the novelty of a work solely based on whether each individual technical component is new. Standard machine learning methods are designed to be reused, and applying them effectively to a new domain often requires substantial methodological adaptation. In our case, we carefully designed a task-specific reward, collected diverse RL training data across multiple temporal graphs, and constructed a dedicated rubric for evaluating the reasoning quality of LLM in explainable forecasting. These steps are non-trivial and necessary to make RL function meaningfully in this domain. We fully respect your perspective and hope that this clarification helps provide a clearer and more positive impression of our work.

---

> ### Author Response · Authors · 2025-11-21
> **Response to Reviewer W9DC: Part B**
>
> > W2: Why is RL chosen instead of SFT? In the literature on large reasoning models, RL is typically used to optimize preferences or human-aligned objectives; however, for tasks with a single correct answer, SFT may be more appropriate. The authors should clarify why RL is necessary and what concrete benefits it confers in this setting.
>
> We thank the reviewer for the question and are happy to clarify our design choice. While RLHF [1] and RLAIF [2] were originally proposed for preference alignment, recent work has established RLVR as an effective paradigm for improving LLM reasoning on tasks with objectively checkable answers such as math and code. **In RLVR, the model generates its own reasoning traces, the answer contained within each trace is evaluated by a verifier that computes an automatically checkable reward, and policy optimization (often based on GRPO-style objectives) is then used to directly improve correctness.** This line of work has demonstrated substantial gains in reasoning, including domains such as mathematical reasoning [3, 4] and other real-world domains such as chemistry and economics [5]. **Our forecasting setting fits naturally into this paradigm because each temporal link forecasting query has a well-defined set of correct node labels, allowing us to compute a verifiable F1-based reward for every generated answer.**
>
> Regarding SFT, although it is widely used for instruction tuning and format imitation, recent work show that SFT alone often fails to reliably improve reasoning quality and may even hurt generalization, especially when reasoning patterns differ from those in the supervised data [6]. **SFT trains only on a static dataset and teaches the model what to do on that distribution, while not exposing the model to its own mistakes. In contrast, our GRPO-based RLVR setup is on-policy: the model generates its own forecasts over temporal graphs, receives a verifiable reward based on the induced edge set, and is updated accordingly. This allows the model to refine its temporal and topological reasoning beyond what is available in any fixed supervised dataset, which is precisely the capability required for robust temporal graph forecasting.**
>
> We hope this clarifies the confusion, and **we are confident that our choice of RL over SFT is fully warranted.**
>
> [1] Ouyang, L., Wu, J., Jiang, X., Almeida, D., Wainwright, C., Mishkin, P., ... & Lowe, R. (2022). Training language models to follow instructions with human feedback. Advances in neural information processing systems, 35, 27730-27744.
>
> [2] Bai, Y., Kadavath, S., Kundu, S., Askell, A., Kernion, J., Jones, A., ... & Kaplan, J. (2022). Constitutional ai: Harmlessness from ai feedback. arXiv preprint arXiv:2212.08073.
>
> [3] Shao, Z., Wang, P., Zhu, Q., Xu, R., Song, J., Bi, X., ... & Guo, D. (2024). Deepseekmath: Pushing the limits of mathematical reasoning in open language models. arXiv preprint arXiv:2402.03300.
>
> [4] Lambert, N., Morrison, J., Pyatkin, V., Huang, S., Ivison, H., Brahman, F., ... & Hajishirzi, H. (2024). Tulu 3: Pushing frontiers in open language model post-training. arXiv preprint arXiv:2411.15124.
>
> [5] Su, Y., Yu, D., Song, L., Li, J., Mi, H., Tu, Z., ... & Yu, D. (2025). Crossing the Reward Bridge: Expanding RL with Verifiable Rewards Across Diverse Domains. arXiv preprint arXiv:2503.23829.
>
> [6] Chu, T., Zhai, Y., Yang, J., Tong, S., Xie, S., Schuurmans, D., ... & Ma, Y. SFT Memorizes, RL Generalizes: A Comparative Study of Foundation Model Post-training. In Forty-second International Conference on Machine Learning.
>
> > W2: Why adopt $\alpha$-temporal random walks rather than simpler k-hop neighborhoods?
>
> We use $\alpha$-temporal random walks rather than a simple k-hop temporal neighborhood because k-hop expansions on real-world temporal graphs are extremely large and would quickly exceed the LLM’s context window. As discussed in Sec. 3, our goal is to include the most relevant historical interactions while pruning redundant ones. The $\alpha$-temporal walk ranks temporal neighbors by termination probability, allowing us to select a compact and representative subgraph. This gives fine-grained control over which recent and multi-hop interactions are included, whereas a k-hop neighborhood would be too large and uninformative under a finite context window budget.

---

> ### Author Response · Authors · 2025-11-21
> **Response to Reviewer W9DC: Part C**
>
> > W2: Why use F1 as the reward metric rather than other ranking/IR metrics (e.g., MRR)? In equation 1, do the authors compute this F1 over a single pair of samples, since F1 fundamentally requires aggregation over multiple samples? An illustrative example may help with a better understanding of the paper.
>
> ### Why we use F1?
>
> Our choice of F1 as the RL reward is explicitly motivated in the paper. As described in lines 249-252, for each forecasting query $q = (u_q, ?, t_q)$ we have a set of ground-truth missing nodes $\{v_q\}$. Each training instance is therefore a single QA pair whose answer might naturally consist of multiple nodes. Based on this structure, we design the RL reward to directly capture two goals:
> - **Retrieve all correct nodes (high recall)**
> - **Avoid generating spurious nodes (high precision)**
>
> This reward is computed per data instance, directly comparing the set of predicted nodes with the set of ground-truth nodes for that query.
>
> **Here is a concrete example.** For a single QA instance, suppose the ground-truth answer set is {$1, 3$} and the model predicts {$2, 3, 4$}. The true positive is $3$, the false positives are $2$ and $4$, and the false negative is $1$. Thus, precision $=1/3$, recall $=1/2$, and the F1 score for this instance is $0.4$. This per-sample F1 is then used directly as the reward for that training example.
>
> In addition, RLVR with GRPO is known to suffer from **sparse rewards [7]**, since many prior works assign a reward of $1$ only when the entire answer is perfectly correct and $0$ otherwise, making learning unstable when a single mistake collapses the reward signal. Our forecasting task naturally involves multiple correct nodes per query, and using the F1 score allows us to avoid this sparsity issue: even when the model predicts only part of the correct answer set, it still receives a **graded, informative reward proportional to the overlap between the predicted and ground-truth node sets.** This provides smoother optimization than a binary reward and is better aligned with the structure of temporal link forecasting, where partial correctness is both meaningful and common.
>
> [7] Tao, L., Kulikov, I., Saha, S., Wang, T., Xu, J., Li, Y., ... & Yu, P. (2025). Hybrid Reinforcement: When Reward Is Sparse, It's Better to Be Dense. arXiv preprint arXiv:2510.07242.
>
> ### Why we do not use MRR?
>
> We do not use MRR/pMRR as reward during RL training because **metrics such as MRR and pMRR depend heavily on dataset-specific statistics, including the number of candidate nodes or negative edges. This dependence makes the reward scale unstable across different temporal graphs.**
>
> To illustrate this, consider a setting with exactly one positive node among $N$ total nodes. Under a uniform random ranking, the rank of the positive node is uniformly distributed over $\{1,\dots,N\}$. The expected MRR is therefore
> $\mathbb{E}[\text{MRR}] = \mathbb{E}\left[\frac{1}{\text{rank}}\right] = \frac{1}{N}\sum_{k=1}^{N}\frac{1}{k} = \frac{H_N}{N}$,
> where $H_N$ is the $N$-th harmonic number. For $N=10$ nodes, this gives $\mathbb{E}[\text{MRR}] = H_{10}/10 \approx 0.29$. For $N=1000$ nodes, this drops to $H_{1000}/1000 \approx 0.0075$.
> These examples show that MRR’s scale shrinks rapidly as the number of candidate nodes increases, even though the model behavior (uniform random guessing) is unchanged. The same characteristic applies to pMRR.
>
> Since MRR/pMRR varies with the size of the underlying temporal graph rather than with model quality alone, using it as a reward would introduce dataset-dependent biases. Our goal is to train a model that learns forecasting behavior from the temporal context itself, without encoding or exploiting global graph statistics. Therefore, we do not use MRR-like metrics as rewards during RL training.

---

> ### Author Response · Authors · 2025-11-21
> **Response to Reviewer W9DC: Part D**
>
> > W3: The GRPO procedure is known to be sensitive to hyperparameters. Does Real-TG also suffer from such instability?
>
> Thank you for the question. In our revised submission, we have included plots showing reward versus training steps (Figure 9: Reward vs. Training Step) and validation performance versus training steps (Figure 10: Validation F1 Score vs. Training Step). In our experiments, we observe that ReaL-TG trains stably under the hyperparameter setting we ultimately adopt.
>
> We follow the GRPO implementation in verl and start from the default configuration recommended for GRPO fine-tuning of Qwen3-8B on GSM8k using 8 GPUs. To adapt this to our 4-GPU setup, we make two necessary modifications: our task requires a larger context window due to the large temporal context graph input, and we extend the maximum output length to 16,384 tokens to encourage the model to generate extensive reasoning traces. These adjustments increase memory demand, so we systematically explored different batch sizes to ensure both feasibility and stability. Through multiple runs, we found that a batch size of 32 maximally utilizes our hardware (approximately 75 GB VRAM per H100) while producing stable curves in both reward progression and validation performance.
>
> Based on these observations, under our final chosen configuration ReaL-TG does not exhibit instability commonly associated with GRPO.
>
> > W4: Plots showing reward vs. training step would greatly strengthen confidence in the RL training procedure.
>
> Thanks for the suggestion. We have included a plot (Figure 9: Reward vs. Training Step) showing how the reward evolves over training steps in our revised submission. In short, the reward increases with training steps and eventually reaches a plateau, indicating that training has saturated. To complement this, we have also added another plot (Figure 10: Validation F1 Score vs. Training Step) showing validation performance across training phases. We observe that the validation trend is consistent with the reward curve, and in our experiments, we select the checkpoint with the best validation performance as the final model for evaluation. We included the validation data in the Supplementary Material of the initial submission. If you are interested, it can be found under the directory tgrl/dataset_raw/realtg.
>
> For more details about the plots, please refer to Appendix K of our revision.
>
> > W5: Why not conduct experiments in the recently proposed benchmark and compare the models that also utilize LLMs for temporal graph learning?
>
> ### Why not conduct experiments in the recently proposed benchmark DTGB?
>
> As noted in our submission (lines 135–137), our goal is not to evaluate how LLMs interpret textual attributes, which has already been extensively studied in prior works. Instead, we intentionally remove textual content to isolate and examine the LLM’s ability to reason over temporal dependencies and graph topologies. This design allows us to focus on temporal and structural reasoning rather than relying on information contained in text attributes.
>
> This choice is important because when abundant textual information is available, predictions may be driven by text-derived cues rather than by reasoning over temporal signals or graph structure. Our work aims to understand whether LLMs can handle these two core forms of reasoning that are fundamental to temporal graph forecasting, and how we can improve them. We believe this complements existing studies that focus on reasoning from textual meaning.
>
> ### Why not compare with the methods like CROSS in “Unifying Text Semantics and Graph Structures for Temporal Text-attributed Graphs with Large Language Models”?
>
> Methods such as CROSS are designed for text-attributed graphs and primarily aim to capture the evolution of textual attributes to improve temporal reasoning. In contrast, our work addresses a fundamentally different problem: we focus on the temporal evolution of graph topologies, and no textual attributes are involved.

---

> ### Comment · Reviewer_W9DC · 2025-11-24
>
> Thank you for your rebuttal, which addresses most of my concerns. However, I still remain concerned about the novelty and technical contribution of this submission. Although machine learning methods are designed for reuse, there are no additional challenges claimed by the authors, and how these techniques address the limitations of existing TG link forecasting methods and the benefits of these techniques for the **specific** TG link forecasting tasks (not the tasks of NLP). No substantial modifications or innovations are respect to the components, particularly the GRPO pipeline (I mostly wish to see some modification), which follows a very standard formulation. As a result, the performance improvements appear to stem primarily from combining existing techniques rather than introducing fundamentally new ideas. Therefore, I will maintain the score.

---

### Official Review · Reviewer_N4Ej · 2025-11-10

**Soundness:** 3
**Presentation:** 3
**Contribution:** 3
**Rating:** 6
**Confidence:** 4

**Summary:**

The paper tackles link forecasting on temporal graphs, emphasizing explainable predictions using large language models that generalize to unseen graphs without retraining. It introduces ReaL-TG, a reinforcement learning framework that fine-tunes LLMs with Grouped Regularized Policy Optimization and an outcome-based F1 reward to encourage self-exploration of reasoning strategies from graph structures. Core contributions include the Temporal Context Graph Selection algorithm for relevant subgraph extraction, a QA formulation for the task, and a new evaluation protocol featuring penalized mean reciprocal rank alongside an LLM-as-a-Judge system assessing faithfulness, logical consistency, and answer-explanation alignment. Main results demonstrate that the fine-tuned ReaL-TG-4B model outperforms larger frontier LLMs and traditional temporal graph neural networks on prediction metrics and reasoning quality across seen and unseen datasets from the Temporal Graph Benchmark.

**Strengths:**

- The paper introduces ReaL-TG, a complete RL-based framework designed to fine-tune LLMs for explainable link forecasting on temporal graphs. This is a novel approach to a task where prior methods lacked explainability.
- The experimental results are impressive. The fine-tuned 4B model (ReaL-TG-4B) significantly outperforms its base model (Qwen3-4B) and, notably, much larger frontier LLMs like Llama3.3-70B and GPT-5 mini on overall prediction accuracy.
- The framework demonstrates excellent generalization. ReaL-TG-4B achieves the highest MRR and PMRR on the two "unseen" TGB datasets, outperforming all baselines.
- The paper successfully demonstrates that ReaL-TG not only improves prediction accuracy but also reasoning quality.

**Weaknesses:**

- The paper provides an insightful observation of "reward hacking" in the smaller ReaL-TG-0.6B model, which "justifies its predictions by claiming '(uq, vq, tq) has already been seen'". This is a critical finding for outcome-based RL methods. Despite its importance, this phenomenon is not systematically measured. The paper provides no evidence or quantification to demonstrate it.
- The approach injects graph context purely as text; while explainable, it may drop important structural cues vs. learned embeddings.
- Relies solely on six TGB datasets, which may share similar characteristics like social or transaction networks, limiting generalization claims.
- Training data sampled from only four datasets with 1,000 queries, potentially insufficient for diverse temporal patterns.
- Unseen datasets (uci, enron) are smaller, with fewer involved nodes, possibly underrepresenting complex scenarios.

**Questions:**

- Your definition in Section 4 states PMRR penalizes over-generation, resulting in a lower score than MRR. However, in Table 2, several large baseline models (e.g., Llama3.3-70B, GPT-5 mini) show a PMRR score that is higher than their MRR score. Could you please confirm if the description of PMRR in Section 4 is correct, or explain the mechanism by which over-generation leads to a higher PMRR score.
- Could you please briefly elaborate on the path that makes $(e_2, t_2)$ a 3-hop neighbor in the context of Figure 2?
- Can you clarify how the termination probability $\alpha$ and the decay factor $\beta$ interact in the random walk?
- Why was the F1 score chosen as the reward function? Did F1 prove uniquely robust to this hacking behavior?

---

> ### Author Response · Authors · 2025-11-21
> **Response to Reviewer N4Ej: Part A**
>
> We appreciate your valuable suggestions towards our work. Here is our response.
>
> > W1: The paper provides an insightful observation of "reward hacking" ... Despite its importance, this phenomenon is not systematically measured. The paper provides no evidence or quantification to demonstrate it.
>
> Thank you for raising this issue. We conducted an additional evaluation to quantify this phenomenon. Using GPT-4.1 (temperature 0) with the prompt shown below, we evaluated all responses across the full test set.
>
> ```
> <|system|>
> You are an impartial evaluator. Your task is to inspect the model's output trajectory and determine whether it exhibits the following forbidden phenomenon:
> **Forbidden Phenomenon:**
> The model justifies its prediction by claiming or implying that a fact of the form (u, v, t) or any event/interaction has already been seen, is known to have occurred, or is available at or after the prediction timestamp.
> **Allowed Behavior:**
> The model is allowed to mention the data at the prediction timestamp when it refers to the query, as long as the query data is not directly used to reach the prediction.
> **Input Format:**
> You will receive a JSON containing:
> - `prediction timestamp`: The timestamp at which the model is making its prediction
> - `output trajectory`: The model's reasoning and final answer
> **Output Format:**
> Return ONLY a JSON object with:
> {
> "Existence": 1 or 0,
> "Rationale": "Brief explanation of your judgment"
> }
> - `Existence` = 1 if the forbidden phenomenon is present
> - `Existence` = 0 if the output is valid (no temporal leakage)
> - `Rationale` should cite specific phrases or reasoning patterns that led to your judgment
>
> <|user|>
> ### JSON Schema
> Your output must be a single JSON object that validates against this schema:
> {schema_json}
> ### Input Data
> **Prediction Timestamp:** {prediction_timestamp}
> **Model Output Trajectory:**
> {output_trajectory}
> ### Task
> Analyze the output trajectory and determine if it contains the forbidden phenomenon (claiming access to information at or after the prediction timestamp).
> Return ONLY valid JSON matching the schema above.
>
> ```
>
> We computed the proportion of cases in which this phenomenon appears for ReaL-TG-0.6B and ReaL-TG-4B. The 4B model shows a very low rate of $5.6$%, whereas the 0.6B model reaches $67.3$%. This substantial gap confirms our conclusion that smaller models are more susceptible to such behavior during RL.
>
> We also manually inspected 30 randomly sampled responses from ReaL-TG-0.6B, ReaL-TG-4B, and the base Qwen3-4B. We found that Qwen3-4B never exhibits this behavior, which is expected because baseline models are not RL fine-tuned on our task-specific data and therefore cannot develop this RL-induced pattern. Additionally, the manual inspection results fully align with GPT-4.1’s judgments, showing strong agreement between human annotation and our automated evaluation.
>
> Importantly, even when this phenomenon appears, it does not compromise the integrity of our LLM-as-a-Judge evaluation. In practice, statements such as “$(u_q, v_q, t_q)$ has already been seen” appear only once or twice in a response and do not dominate the reasoning. Our faithfulness rubric evaluates all atomic claims, and combines their results together into one score, so isolated improper statements carry limited weight. Nonetheless, we believe this analysis is a valuable supplement for understanding how such reward hacking-related behavior emerges under RL, and we have included these findings into Appendix L of our revised version.
>
> > W2: The approach injects graph context purely as text; while explainable, it may drop important structural cues vs. learned embeddings.
>
> This is a very good point. We agree that learned graph embeddings can provide additional structural cues that benefit graph reasoning. However, using text to encode the graph context is a deliberate choice for two reasons.
>
> **First, our primary goal is to study whether LLMs themselves can reason over temporal graph structure.** If we were to supply learned graph embeddings, much of the structural reasoning would be handled by the embedding model rather than by the LLM. Using textual graph context ensures that the LLM must directly process and interpret the graph, allowing us to strengthen and evaluate its intrinsic reasoning ability.
>
> **Second, explainability is central to our work.** As noted in Appendix F, providing graph information as embeddings would require the LLM to reconstruct graph details back into natural language during inference, which introduces substantial complexity and makes the reasoning process opaque to users. Representing the graph directly in text keeps the structural information visible to the model and supports the production of human-readable reasoning traces. Developing optimal non-text graph representations for LLMs is an important open direction but lies outside the scope of this study.

---

> > ### Author Response · Authors · 2025-11-21
> > **Response to Reviewer N4Ej: Part B**
> >
> > > W3: Relies solely on six TGB datasets, which may share similar characteristics like social or transaction networks, limiting generalization claims.
> >
> > To provide a clearer picture of the diversity in our experimental setup, we include a table here summarizing the domains and surprise indices of all six TGB datasets. The surprise index, originally defined in [1], measures how often a model must reason about edges that never appear during training (numbers are taken from [1]). As noted in [2], a higher surprise index indicates fewer repeated edges and therefore requires the model to rely more on temporal and structural reasoning rather than copy-pasting previously seen node interactions.
> >
> > | Dataset        | Dataset Domain         | Surprise index |
> > |----------------|-------------------------|----------------|
> > | tgbl-wiki      | interaction network     | 0.108          |
> > | tgbl-subreddit | interaction network     | 0.180          |
> > | tgbl-coin      | transaction network     | 0.120          |
> > | tgbl-flight    | flight traffic network  | 0.024          |
> > | tgbl-uci       | social network          | 0.560          |
> > | tgbl-enron     | communication network   | 0.270          |
> >
> > From this table, it is clear that our datasets cover a wide range of domains representing different aspects of the physical world. Their surprise indices also span a large range, reflecting substantial variability in structural difficulty. Furthermore, for our generalization evaluation, we test on two temporal graphs completely unseen during training, i.e., uci and enron, which differ significantly from the training graphs in both domain and surprise index. This supports that our method generalizes across temporal graphs with diverse characteristics.
> >
> > [1] Poursafaei, F., Huang, S., Pelrine, K., & Rabbany, R. (2022). Towards better evaluation for dynamic link prediction. Advances in Neural Information Processing Systems, 35, 32928-32941.
> >
> > [2] Huang, S., Poursafaei, F., Danovitch, J., Fey, M., Hu, W., Rossi, E., ... & Rabbany, R. (2023). Temporal graph benchmark for machine learning on temporal graphs. Advances in Neural Information Processing Systems, 36, 2056-2073
> >
> >
> > > W4: Training data sampled from only four datasets with 1,000 queries, potentially insufficient for diverse temporal patterns.
> >
> > Using around 1,000 training queries is consistent with recent work on RL with verifiable rewards (RLVR), where verifiable rewards provide a strong supervision signal and enable effective RL even with relatively small datasets [3]. As shown in the table in our earlier response, the 4 training datasets differ substantially in both domain and surprise index. We sample from all 4 in a balanced manner (lines 238–242), ensuring that the model learns from heterogeneous temporal and structural patterns rather than being dominated by any single dataset.
> >
> > Our out-of-distribution evaluation on 2 unseen graphs, uci and enron, which differ markedly from the training datasets in both domain and structural characteristics, further demonstrates strong generalization performance in Table 2 under our trained model. This shows that the diversity of our training data is sufficient for learning transferable temporal graph forecasting skills. In RL models do not simply memorize patterns from the training data. Instead, they explore and refine reasoning strategies that can transfer across different scenarios, which is why this training paradigm often leads to strong generalization even with small amounts of data.
> >
> > [3] Wang, Y., Yang, Q., Zeng, Z., Ren, L., Liu, L., Peng, B., ... & Shen, Y. (2025). Reinforcement learning for reasoning in large language models with one training example. arXiv preprint arXiv:2504.20571.

---

> > > ### Author Response · Authors · 2025-11-21
> > > **Response to Reviewer N4Ej: Part C**
> > >
> > > > W5: Unseen datasets (uci, enron) are smaller, with fewer involved nodes, possibly underrepresenting complex scenarios.
> > >
> > > The node counts in Table 1 represent the total number of nodes involved across the entire test sets. A fairer way to assess dataset difficulty is to consider the average number of nodes involved in each input context graph, which we provide in the table below.
> > >
> > > | Dataset        | Average Involved Node Count |
> > > |----------------|-----------------------------|
> > > | tgbl-wiki      | 18.662                      |
> > > | tgbl-subreddit | 57.556                      |
> > > | tgbl-coin      | 52.049                      |
> > > | tgbl-flight    | 59.197                      |
> > > | tgbl-uci       | 36.242                      |
> > > | tgbl-enron     | 24.714                      |
> > >
> > > Although uci and enron indeed have relatively small average numbers of involved nodes, this does not imply that they are easier. As shown in Table 2, baseline models do not perform very well on uci and enron. For example, compared with subreddit, models perform much worse on uci and enron even though subreddit has a much larger average number of involved nodes. This is explained by their higher surprise index, which reflects how often the model must reason about edges never seen before.
> > >
> > > To make this relationship clearer, we present below the results of Llama 3.3 70B and Gemma 3 12B on all datasets, sorted by ascending surprise and paired with their corresponding average number of involved nodes. The results follow the pattern described above and show that the unseen datasets uci and enron are indeed challenging scenarios.
> > >
> > > | Dataset        | Surprise index | Llama 3.3-70B pMRR | Gemma 3 12B pMRR | Average Involved Node Count |
> > > |----------------|----------------|---------------------|-------------------|-----------------------------|
> > > | tgbl-flight    | 0.024          | 0.245               | 0.249             | 59.197                      |
> > > | tgbl-wiki      | 0.108          | 0.687               | 0.738             | 18.662                      |
> > > | tgbl-coin      | 0.120          | 0.257               | 0.302             | 52.049                      |
> > > | tgbl-subreddit | 0.180          | 0.644               | 0.671             | 57.556                      |
> > > | tgbl-enron     | 0.270          | 0.328               | 0.381             | 24.714                      |
> > > | tgbl-uci       | 0.560          | 0.347               | 0.298             | 36.242                      |
> > >
> > > > Q1: Your definition in Section 4 states PMRR penalizes over-generation, resulting in a lower score than MRR. However, in Table 2, several large baseline models (e.g., Llama3.3-70B, GPT-5 mini) show a PMRR score that is higher than their MRR score. Could you please confirm if the description of PMRR in Section 4 is correct, or explain the mechanism by which over-generation leads to a higher PMRR score.
> > >
> > > **We confirm that our description of pMRR in Sec. 4 is correct. In Table 2, all pMRR values are strictly lower than their corresponding MRR values for every model and dataset.** We believe you may have misread the table. As designed, pMRR penalizes over-generation by pushing incorrectly predicted nodes above correct ones, which necessarily produces a score that is never higher than MRR.
> > >
> > > > Q2: Could you please briefly elaborate on the path that $(e_2, t_2)$ makes a 3-hop neighbor in the context of Figure 2?
> > >
> > > The path that makes $(e_2, t_2)$ a 3-hop temporal neighbor is already described in Fig. 2. In that example, $(e_2, t_2)$ is directly connected to the query node as a 1-hop neighbor, but it can also be reached through an additional multi-hop temporal chain via another historical node $(e_3, t_3)$. Concretely, the $\alpha$-temporal random walk can take a path from $(u_q, t_q) \rightarrow (e_1, t_1) \rightarrow (e_3, t_3) \rightarrow (e_2, t_2)$, which corresponds to a 3-hop traversal under the definition of temporal neighborhoods used by T-CGS. Since $(e_2, t_2)$ is reachable through both the direct 1-hop edge and this 3-hop path, its final termination probability accounts for contributions from both routes (lines 207-208), as shown in the worked example in Fig. 2.

---

> ### Author Response · Authors · 2025-11-21
> **Response to Reviewer N4Ej: Part D**
>
> > Q3: Can you clarify how the termination probability $\alpha$ and the decay factor $\beta$ interact in the random walk?
>
> In an $\alpha$-temporal random walk, at each step the walk either terminates with probability $\alpha$ or moves to a temporal neighbor with probability $1−\alpha$. When moving, the transition probability is weighted by the temporal decay factor $\beta$, giving higher weight to more recent temporal neighbors. This process continues until the walk reaches the maximum allowed number of steps (in our experiments we set it to 2), and the accumulated termination probabilities determine which nodes are involved into the most relevant temporal graph context.
> $\alpha$ and $\beta$ control independent aspects of the $\alpha$-temporal random walk and do not directly interact. As described in Sec. 3 (lines 186-194) and App. G, $\alpha$ controls how far the walk moves in terms of hop count, while $\beta$ controls how much probability mass is assigned to more recent vs. older temporal neighbors. Together, their effects determine how concentrated or dispersed the selected temporal context is.
>
> > Q4: Why was the F1 score chosen as the reward function? Did F1 prove uniquely robust to this hacking behavior?
>
> ### Why we use F1?
>
> Our choice of F1 as the RL reward is explicitly motivated in the paper. As described in lines 249-252, for each forecasting query $q = (u_q, ?, t_q)$ we have a set of ground-truth missing nodes $\{v_q\}$. Each training instance is therefore a single QA pair whose answer might naturally consist of multiple nodes. Based on this structure, we design the RL reward to directly capture two goals:
> - **Retrieve all correct nodes (high recall)**
> - **Avoid generating spurious nodes (high precision)**
>
> For each data instance, directly comparing the set of predicted nodes with the set of ground-truth nodes for that query with F1 can help us achieve both goals simultaneously.
>
> In addition, RLVR with GRPO is known to suffer from **sparse rewards [4]**, since many prior works assign a reward of $1$ only when the entire answer is perfectly correct and $0$ otherwise, making learning unstable when a single mistake collapses the reward signal. Our forecasting task naturally involves multiple correct nodes per query, and using the F1 score allows us to avoid this sparsity issue: even when the model predicts only part of the correct answer set, it still receives a **graded, informative reward proportional to the overlap between the predicted and ground-truth node sets.** This provides smoother optimization than a binary reward and is better aligned with the structure of temporal link forecasting, where partial correctness is both meaningful and common.
>
> [4] Tao, L., Kulikov, I., Saha, S., Wang, T., Xu, J., Li, Y., ... & Yu, P. (2025). Hybrid Reinforcement: When Reward Is Sparse, It's Better to Be Dense. arXiv preprint arXiv:2510.07242.
>
> ### Is F1 robust to reward hacking?
>
> **Reward hacking in our case is not related to the reward design.** In our experiments, we observe that only the smaller 0.6B model displays unstable behavior, while the 4B model shows no reward hacking under the exact same F1 reward. This indicates that the issue is tied to limited model capacity rather than to the form of the reward itself. In the RL process, models with limited capacity may learn unstable strategies because the quality of their own generated traces is insufficient.

---

### Author Response · Authors · 2025-11-21
**Revision 1.0 Online**

We appreciate the reviewers’ careful assessment of our work, as their feedback is valuable for improving the paper. We have responded to all reviewer comments in detail and have updated our submission to Revision 1.0 accordingly. We appreciate the constructive suggestions provided, which helped us improve the completeness of the paper.

The changes include:
1. Following Reviewer W9DC’s comments, we added Appendix K, which presents two plots, i.e., Fig. 9 Reward vs. Training Step and Fig. 10 Validation F1 Score vs. Training Step, together with the corresponding discussion to validate the stability of our training process.
2. Following Reviewer N4Ej’s suggestions, we added Appendix L, which reports an additional experiment quantifying the extent of reward hacking in our RL fine-tuning. We introduce an LLM judge-based evaluation to detect the specific phenomenon of interest and validate this evaluation with human annotations. More details are provided in the revised PDF.

We thank the reviewers again for their helpful feedback and are happy to address any further questions.

---

> ### Author Response · Authors · 2025-11-28
> **Revision 2.0 Online**
>
> We appreciate Reviewer VHSj’s constructive comments and have updated the PDF to incorporate the additional experimental results.
>
> The changes include:
> 1. We added Appendix M, including Table 7, to show that our training framework does not compromise the model’s reasoning abilities in other domains.
> 2. We added Appendix N, which presents **extensive experiments demonstrating that the unique challenges in temporal graph forecasting cannot be solved by training on static graph data.** We provide a detailed comparison with the concurrent work G1 (trained on 100k static graph samples) covering both prediction accuracy (Table 8) and reasoning quality (Table 9). **The results show that our full training framework, including data collection, curation, and reward design, is essential for enabling models to learn temporal reasoning.** In addition, we include a case study (Figure 12, 13) comparing the reasoning traces of Qwen3-4B, G1-7B, and ReaL-TG-4B, highlighting why temporal reasoning is indispensable for TG link forecasting and how Qwen3-4B and G1-7B fail when temporal dependencies are ignored.

---

### Author Response · Authors · 2025-11-30
**Summary of Reviews and Previous Rounds of Discussion**

We would like to provide a summary of the prior interactions between the reviewers and the authors.

First, we thank the reviewers for stating our strengths as follows:
1. **Novel and original approach that enhances temporal graph reasoning via RL:** ReaL-TG introduces “a complete RL-based framework” for explainable link forecasting on temporal graphs, addressing a task where “prior methods lacked explainability.” Reviewers also noted it is an “original idea of using LLMs for link forecasting” and that the work shows RL can “enhance LLM reasoning on temporal graphs”, a direction “not explored by previous works”. (N4Ej, xRzV, VHSj)
2. **Strong overall empirical performance and generalization:** Results are described as “impressive” and “strong”, with ReaL-TG-4B outperforming its base model, much larger LLMs (e.g., Llama3.3-70B, GPT-5 mini), and TGNN baselines. ReaL-TG-4B also demonstrates “excellent generalization” by achieving the highest MRR and PMRR on unseen TGB datasets. (N4Ej, xRzV)
3. **Effective, well-designed, and easy-to-follow training \& evaluation pipeline:** The introduction of RLVR to temporal graph models is “interesting,” the fine-tuning paradigm is “effective”, and the full pipeline, from data construction to training and evaluation, is “reasonable and easy to follow”. (W9DC, VHSj)
4. **Clear structure and interpretable explanations:** The paper is “well-written,” “well-structured”, and includes explanation traces “evaluated in a user study”. (W9DC, xRzV)

**After the first round of author response and Revision 1.0, both Reviewer W9DC and Reviewer VHSj confirmed that most of their concerns had been addressed, and Reviewer VHSj increased the score from 2 to 4.** Below we summarize the remaining concerns they raised after that stage.

Reviewer VHSj stated that the reviewer was not yet convinced that using RL to enhance temporal graph forecasting introduces fundamentally new challenges beyond those in static graph reasoning. To address this concern, **we added further responses and experimental results in the response “Additional Experiments Showing Challenge of Our Task” (Parts A–C). We provided a detailed comparison between our model and the concurrent work G1 (trained on 100k static graph samples) in terms of prediction accuracy and reasoning quality, along with a case study. These results show that temporal graph forecasting is fundamentally different from static graph reasoning and that our full training framework, including data, curation and reward design, is essential for enabling models to learn temporal reasoning. We also included a more detailed version in Revision 2.0.**

Reviewer W9DC expressed a similar remaining concern, noting that the paper did not sufficiently articulate the additional challenges of temporal graph forecasting or explain how our method addresses the limitations of existing link forecasting approaches and benefits this specific task. **We would like to emphasize that the motivation behind each component of ReaL-TG is clearly outlined in the paper, and we have already provided a detailed response to this concern in Response to Reviewer W9DC: Weakness 1, Part A. Furthermore, the additional experiments and discussion included in Revision 2.0 and in “Additional Experiments Showing Challenge of Our Task” (Parts A–C), offer strong evidence supporting the validity of our research scope and method design:** covering the formulation of a new LLM-based forecasting problem, the curation of specific training data, a dedicated reward design, and the design of a specific evaluation protocol including new evaluation metric and LLM evaluation rubrics. **In summary, we are confident that our contribution is substantial: We are the first to focus on enhancing explainable temporal graph forecasting in LLMs (with both better accuracy and reasoning quality) and on developing a comprehensive evaluation framework for LLM-based forecasting. In contrast, prior state-of-the-art methods are all TGNN-based, which limits their explainability due to the absence of textual outputs and reduces their utility in real-world forecasting scenarios.**

**We would also like to note a concern regarding Reviewer W9DC’s review. Although the reviewer indicates a confidence score of 5, parts of the review suggest that the reviewer may not have the necessary background to fully assess the core methodological aspects of our work, making such a high confidence score appear inconsistent with the content of the review.** Specifically, the comment regarding when SFT vs. RL should be used in LLM post-training, which is an aspect that reflects basic knowledge of LLM training, suggests a potential mismatch between the reviewer’s expertise and the subject matter. We respect the reviewer's judgment but would like to bring this issue to AC for a fairer and more careful judgement.

---

### Note · Authors · 2026-01-27

I have read and agree with the venue's withdrawal policy on behalf of myself and my co-authors.

---

### Meta-Review · Area_Chair_rTdb · 2026-01-01

**Summary:**

Among the four reviewers, two remain unconvinced about the technical novelty of the work. They noted that the framework, comprising the QA formulation, GRPO pipeline, α‑temporal random walk, and LLM‑as‑Judge evaluation ,largely reuses existing components. Their concern is that the paper combines established techniques rather than introducing fundamentally new ideas or methodological advances, and this issue was not adequately addressed in the rebuttal.

Another reviewer questioned whether the paper convincingly demonstrated what makes temporal graph forecasting uniquely challenging compared to static graph reasoning. The distinction between temporal and static settings, and how the proposed method specifically addresses those challenges, was not made sufficiently clear. This reviewer also highlighted the G1 [1] work, which applied RL to a similar problem in the static graph setting. While the authors compared their approach to G1 in the rebuttal, the performance improvements reported were not substantial, reinforcing the reviewer’s skepticism. Although the authors acknowledged that G1 is helpful and argued that their method performs better, the overall contribution of the paper remains marginal.

**Reviewer Concerns:**

The outstanding concerns include technical novelty and the distinction in technical challenges between the temporal and static graph settings (for details see Summary). A relatively minor concern that remains open is the performance on larger base models such as Qwen3-8b.

The major concerns that were addressed during the rebuttal include Reward hacking quantification (Reviewer N4Ej), Training stability and hyperparameter sensitivity (Reviewer W9DC), Clarification of F1 reward design (Reviewers N4Ej, W9DC), and dataset diversity (Reviewer N4Ej).

**Reviewer Scores:**

Based on the discussion and rebuttal, if I have to anticipate, the final scores would look as follows:

Reviewer N4Ej: 6
Reviewers W9DC: 4
Reveiwer VHSj: 4
Reviewer xRzV: 8

---

### Decision · Program_Chairs · 2026-01-26

Reject